# Association between eating behavior and the immediate neural activity caused by viewing food images presented in and out of awareness: A magnetoencephalography study

**Rika Ishida, Akira Ishii** **\*, Takashi Matsuo, Takayuki Minami, Takahiro Yoshikawa**

Department of Sports Medicine, Osaka City University Graduate School of Medicine, Osaka, Japan

\* a.ishii@med.osaka-cu.ac.jp

## Abstract

Obesity is a serious health problem in modern society. Considering the fact that the outcomes of treatments targeting appetitive behavior are suboptimal, one potential reason proposed for these poor outcomes is that appetitive behavior is driven more by unconscious decision-making processes than by the conscious ones targeted by traditional behavioral treatments. In this study, we aimed to investigate both the conscious and unconscious decision-making processes related to eating behavior, and to examine whether an interaction related to eating behavior exists between conscious and unconscious neural processes. The study was conducted on healthy male volunteers who viewed pictures of food and non-food items presented both above and below the awareness threshold. The oscillatory brain activity affected by viewing the pictures was assessed by magnetoencephalography. A visual backward masking procedure was used to present the pictures out of awareness. Neural activity corresponding to the interactions between sessions (i.e., food or non-food) and conditions (i.e., visible or invisible) was observed in left Brodmann's areas 45 and 47 in the high-gamma (60–200 Hz) frequency range. The interactions were associated with eating behavior indices such as emotional eating and cognitive restraint, suggesting that conscious and unconscious neural processes are differently involved in eating behavior. These findings provide valuable clues for devising methods to assess conscious and unconscious appetite regulation in individuals with normal or abnormal eating behavior.

## Introduction

Obesity is a serious health problem in modern society: The prevalence of overweight individuals and obesity are increasing worldwide [1]. Overweight and obesity cause a wide variety of health problems, including type 2 diabetes, dyslipidemia, hypertension, coronary heart disease, and certain kinds of cancer, such as colon cancer [2–4], which has a major effect on the

**Data Availability Statement:** The Ethics Committee of Osaka City University which approved the protocol of our present study does

not allow public sharing of the raw MEG and MRI data since the data contain potentially sensitive information. The values behind the means and standard deviations, which are used to build figures, and the code for experimental task can be received by e-mail upon reasonable request. Non-author point of contact: The Ethics Committee of Osaka City University (ethics@med.osaka-cu.ac.jp).

**Funding:** "AI: JSPS KAKENHI Grant No. 16H03248 and TY: JSPS KAKENHI Grant No. 22K11732".

**Competing interests:** The authors have declared that no competing interests exist.

increasing costs of medical care [5–7]. Since body weight depends on the balance between energy expenditure and food intake, behaviors related to food intake are important intervention targets to prevent obesity. It is estimated that even less than 0.5% of caloric intake over energy expenditure can lead to weight gain [8, 9].

Educational and/or behavioral interventions for addressing obesity are partially successful, with nearly half of treated patients returning to pre-treatment weights within 5 years of intervention completion [10–13]. Considering that the outcomes of treatments targeting appetitive behavior are suboptimal, one potential reason for these poor outcomes is that appetitive behavior is driven more by unconscious decision-making processes than by the conscious decision-making processes targeted by traditional behavioral treatments [14]. In fact, it has been demonstrated that the greater saliva production and the increased rating of hunger were caused by coupling the food-related stimuli presented below the threshold of awareness with positively valenced terms, suggesting that the unconscious processes, which would not evoke eating-related deliberation, can effectively modulate affective and motivational responses caused by food-related stimuli [15, 16].

It has also been reported that neural responses to visual food stimuli reflect some aspects of eating behaviors, and the neural mechanisms related to eating behaviors have been investigated based on neural responses to visual food stimuli [17]. In most of these studies, the visual food stimuli were presented above the threshold of awareness. However, it is of note that there has been a study in which the neural response to visual food stimuli presented below the threshold of awareness was examined using magnetoencephalography (MEG). Visual food stimuli were presented below the threshold of awareness using a backward masking procedure in their study, and the alterations in autonomic nervous activity and the neural response caused by the presentation of the stimuli were examined. Compared with a condition in which mosaic pictures created from original food stimuli were presented so as not to be recognized by the participants, the food stimuli presented so as not to be recognized by them caused the activation of sympathetic nervous activity and alterations in neural activity in Brodmann's areas (BA) 47 and 13, which are related to the alteration of sympathetic nervous activity and the level of cognitive restraint of food intake, respectively [18]. These findings suggest the existence of unconscious processes related to eating behavior. The approach applied in the study that presenting visual food stimuli below the threshold of awareness seems to be a promising method to investigate the neural mechanisms of the unconscious processes related to eating behavior. Although it is important to investigate the interaction between the conscious and unconscious processes related to eating behavior to gain a better understanding of the roles played by conscious and unconscious neural processes in eating behavior, the differences in neural responses between conditions in which visual food stimuli are presented above compared with below the threshold of awareness was not examined in the study.

In this study, we aimed to investigate both the conscious and unconscious processes related to scores for eating behavior assessed by a questionnaire, and to examine whether an interaction associated with the scores for eating behavior exists between the conscious and unconscious neural processes. To achieve this aim, we presented food and non-food pictures both above and below the threshold of awareness and recorded the associated neural activities over the presentations using MEG. We used a visual backward masking procedure [19–21] to present visual food cues below the threshold of awareness. We recorded electrocardiograms (ECGs) over the visual presentations and assessed indices derived from heart rate variability, which reflects both sympathetic and parasympathetic nervous activity. In one MEG study in which food and control images were presented below the threshold of awareness, the inferior frontal gyrus was reported to be involved in the inhibitory control of appetite [18], and in another in which the images were presented above the threshold of awareness, the neural

response correlated to appetitive behavior caused by visual food stimuli was observed as early as 200 ms from the onset of the presentation [22]. In addition, it has been reported that lexical processing and thus the activation of meaning seem to start after around 200 ms of stimulus onset [15, 23, 24], suggesting that the unconscious processing of the presented stimulus is performed within the time window around 0–200 ms from stimulus onset. Therefore, in this study, since we were interested in the neural mechanisms related the regulation of appetite in terms of conscious and unconscious process for eating behavior, we focused on the neural activities in the frontal brain regions immediately after the onset of visual food stimuli (i.e., 0–250 ms). Next, we analyzed the MEG data in the theta (4–8 Hz), alpha (8–13 Hz), beta (13–25 Hz), gamma 1 (25–58 Hz), gamma 2 (62–80 Hz), and high-gamma (62–200 Hz) frequency ranges because several studies have reported that the oscillatory brain activity in the high-gamma frequency range is related to cognitive processes such as attention and conscious perceptions [25–28]. It is of note that the effects caused by viewing the food images may include those of arousal induced by the images (i.e., another arousing stimulus other than food images was not employed as a control stimulus in our present study).

## Materials and methods

### Participants

Thirty-one healthy male volunteers (mean age ± standard deviation [SD], 22.4 ± 2.0 years; body mass index, 22.4 ± 3.0 kg/m$^2$) participated in this study. All participants were confirmed to be right-handed according to the Edinburgh Handedness Inventory [29]. Current cigarette smokers, individuals with a history of mental or neural and/or upper extremity disease, and those taking chronic medicine that affect the activity of the central nervous system were excluded. The study protocol was approved by the Ethics Committee of Osaka City University Graduate School of Medicine (License No. 4151). Each participant provided written informed consent in compliance with the principles of the Declaration of Helsinki and the Ethical Guidelines for Medical and Health Research Involving Human Subjects in Japan (Ministry of Education, Culture, Sports, Science and Technology and Ministry of Health, Labour and Welfare).

### Experimental design

The present study consisted of two conditions (visible and invisible), each of which was performed on a different day. Since we decided to confirm that our participants were not aware of the food and object images in the invisible trials and were aware of the food and object images in the visible trials by asking them whether they recognized images other than the mask image afterward, as described below, the visible and invisible stimuli were not presented in the same session. Each condition had two sessions (food and object). Therefore, this study was conducted in a four-crossover fashion (Fig 1A). The mean interval between days was approximately 1 week. With the exception of water, the participants were instructed to fast from 21:00 on the day before each experimental day (for 12 h before each test), to avoid intense exercise, and to maintain their usual sleeping hours. At the beginning of the experiment on the first day (Day 1), the participants were asked to answer the Japanese version of the revised 21-item version of the Three-Factor Eating Questionnaire (TFEQ) to assess usual eating behaviors [30–32].

During the food and object sessions under the visible and invisible conditions, the participants lay in a supine position on a bed placed in a magnetically shielded room and were asked to look at a visual stimulus presented by a projector (PG-B10S; Sharp, Osaka, Japan) on a screen in front of them.

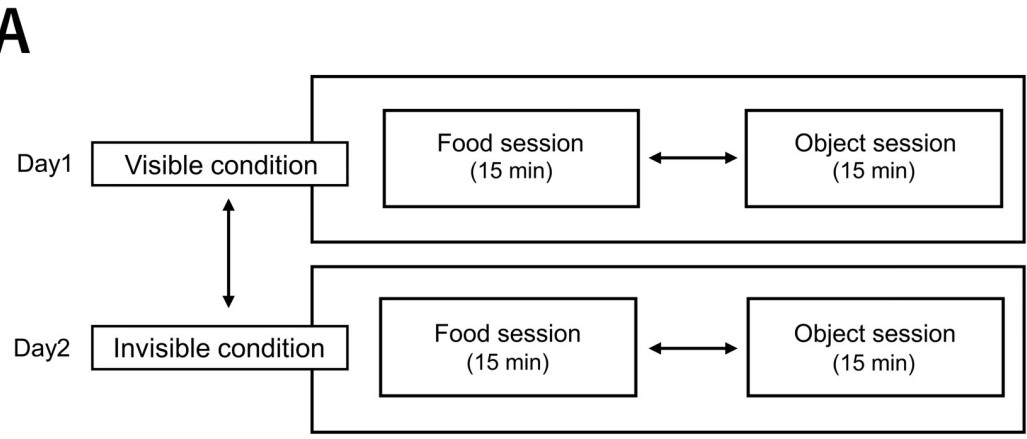

**A**

**B**

**C**

**Fig 1. Experimental procedure.** (A) This study consisted of two conditions (visible and invisible), each of which was performed on a different day. Each condition had two sessions (food and object). (B) The visual stimuli used under the visible condition. The visual stimuli were presented so that the participants could recognize the items (i.e., the food items and objects) in the pictures. (C) The visual stimuli used under the invisible condition. The visual stimuli were presented using a visual backward masking procedure so that the participants could not recognize the items in the pictures.

Under the visible condition, the visual stimuli were presented so that participants were able to recognize the items (i.e., the food items and objects) in the pictures (Fig 1B). Under the invisible condition, the visual stimuli were presented using a visual backward masking procedure [19–21] so that participants could not recognize the items in the pictures (Fig 1C). The visual stimuli used in the food session consisted of a fixation cross (for 1400 ± 200 ms), a food picture (33.4 ms), and a mask picture (1500 ms): The visual presentation in the visible condition was comparable to that in the invisible condition except for the difference of the vertical position of the mask picture. The mask picture was the same for every trial across the food and object sessions in the visible and invisible conditions: The mask picture was created by arranging colored lines and circles and was not created by processing specific picture. This sequence of visual presentations was played 300 times. The set of food pictures used in our present study was identical to those used in our previous studies [18, 22, 33–35]: Twenty pictures of typical modern Japanese food items, including palatable food such as steak, hamburger, pizza, fried rice, and takoyaki (octopus balls), were used as food pictures. Each picture was used 15 times to constitute the 300-picture set. The visual stimuli used in the object session consisted of a fixation cross (for 1400 ± 200 ms), an object picture (33.4 ms), and a mask picture (1500 ms). Twenty pictures of non-food objects were chosen from a freely available database of food and non-food images (http://www.eat.sbg.ac.at/resources/food-pics) [36] so that the color, luminance, size, and complexity of the object pictures matched those of the food pictures. A picture identical to that used as the mask picture was co-presented with the food and object pictures (Fig 1B and 1C). The picture co-presented with the food and object pictures was completely the same as the mask picture (i.e., the actual mask was used). The reason why we added this co-presented mask picture was to attenuate the potential effect caused by the difference of the number of the recognized images between the visible and invisible conditions: The participants would recognize the fixation, the food or object items, and the mask pictures in the visible condition and they would recognize the fixation and the mask picture in the invisible condition if the mask picture was not co-presented with the food or object pictures (i.e., the number of the images recognized by the participants would be three in the visible condition and that would be two in the invisible condition). All mask pictures were the same in every session across the conditions: The position of the mask picture presented on the screen was the same as that of the food and object pictures under the invisible condition, whereas the position of the mask picture presented on the screen was not the same as that of the food and object pictures under the visible condition, so that visual backward masking would not occur. The fixation and the food or object items were always presented near the bottom of the upper half of the screen.

The participants were instructed to look at the fixation cross (i.e., "+") and to continue looking at the same point even after the cross disappeared. The neural activity caused by viewing the visual stimuli in the food and object sessions under the visible and invisible conditions was recorded by MEG.

Just after the experiments under the invisible condition, the participants were asked whether they had recognized any pictures other than the mask pictures via a questionnaire. Just after the experiments under the visible condition, the participants were asked whether they had been able to recognize the food and object pictures during the food and object sessions, respectively, via a questionnaire.

## MEG recording

A 160-channel whole-head type MEG system (MEG version; Yokogawa Electric Corp., Tokyo, Japan) was used to assess the neural activity occurring in the food and object sessions under the visible and invisible conditions. The MEG system used in this study had a magnetic field resolution of 4 fT/Hz$^{1/2}$ in the white noise region. The sensor and reference coils were gradiometers (15.5-mm diameters, 50-mm baseline), and the two coils were 23 mm away. The sampling rate was 1000 Hz. The MEG data were high-pass filtered at 0.3 Hz to exclude the low-frequency component.

## MEG analysis

As a preprocessing step, the MEG data were processed as follows. Magnetic noise from outside the shielded room was eliminated by subtracting the data obtained from the reference coils using specialized software (MEG 160; Yokogawa Electric Corp.). Then, the parts of the MEG data that included artifacts were identified visually and excluded from the analysis. The MEG data were analyzed using a spatial filtering method to identify the changes in oscillatory brain activity that reflected cortical activities induced by the visual stimuli (i.e., the food items and objects) presented under the visible and invisible conditions.

The MEG data were band-pass filtered at 4–8 Hz (i.e., theta band), 8–13 Hz (i.e., alpha band), 13–25 Hz (i.e., beta band), 25–58 Hz (i.e., gamma band 1), 62–80 Hz (i.e., gamma band 2), and 80–200 Hz (i.e., high-gamma band) using a finite impulse response filtering method implemented in Brain Rhythmic Analysis for MEG software (BRAM; Yokogawa Electric Corp.). Then, the locations and intensities of the oscillatory brain activities were estimated using BRAM, which employs a narrow-band adaptive spatial filtering algorithm [37, 38]. The voxel size was set at $5.0 \times 5.0 \times 5.0$ mm. The power of the oscillatory brain activity in each frequency band in the time window of 250 ms after the onset of the presentation of the mask pictures (i.e., 0–250 ms) was calculated relative to that in the time window of –500 to 0 ms from the onset of the presentation of the mask picture (i.e., baseline).

Group analyses of the data produced by the spatial filtering analyses were performed using statistical parametric mapping software (SPM8; Wellcome Department of Cognitive Neurology, London, UK), implemented in MATLAB 2011 (MathWorks, Natick, MA). A magnetic resonance image (MRI) of each participant was transformed into the Montreal Neurological Institute T1-weighted image template, and the parameter used in this transformation was applied to the data produced by the spatial filtering analyses to normalize the MEG data. Smoothing of the anatomically normalized MEG data was performed using a Gaussian kernel of 20 mm (full-width at half-maximum) in the x-, y-, and z-axes.

Individual data were summarized and incorporated into a random-effects model within the frontal brain regions. The estimated parameters were used to create "contrast" images for group analyses. These individual contrast images were used for the analysis of a flexible factorial design with sessions (i.e., food or object) and conditions (i.e., visible or invisible) as within-subject factors. The interaction was calculated for each frequency band. The significance of the interaction was assessed on a voxel-by-voxel basis. The threshold for each analysis was set at $P < 0.00417$ (family-wise error-corrected for multiple comparisons), considering the multiple comparisons among frequencies and the increase or decrease of the oscillatory brain activity compared with baseline (i.e., six frequency bands $\times$ increase or decrease = 12 comparisons). The brain regions that survived the statistical threshold were identified. Localization of the brain regions was executed using WFU_PickAtras, Version 3.0.4 (http://fmri.wfubmc.edu/software/pickatlas) and Talairach Client (Version 2.4.3; http://www.talairach.org/client.html). Small volume correction (i.e., the analyses in the frontal brain regions) was performed using

WFU_PickAtras. The extent to which the neural responses to the pictures in the visible condition was different from those in the invisible condition was assessed by the values calculated as the contrast of the levels of oscillatory brain activity [(food pictures in visible condition + object pictures in invisible condition)–(object pictures in visible condition + food pictures in invisible condition)] for each participant, corresponding to the interactions assessed in the SPM analyses.

## Anatomical MRIs

Anatomical MRIs were obtained for each participant to generate participant-specific MEG source models. The images were obtained using a Philips Achieva 3.0 TX MRI system (Royal Philips Electronics, Eindhoven, the Netherlands). Five MRI-compatible markers (Medtronic Surgical Navigation Technologies, Broomfield, CO) were placed on the scalp (i.e., two markers 10 mm in front of the left and right tragi, one marker 35 mm above the nasion, and two markers 40 mm to either side of the marker above the nasion). The MEG data were overlaid on the MRIs using information obtained from these five markers and the MEG localization coils.

## Electrocardiograms (ECGs)

To assess changes in autonomic nervous activity caused by viewing the food and object pictures presented under the visible and invisible conditions, ECGs were recorded using the EEG system (EEG-1518; Nihon Kohden, Tokyo, Japan) at a sampling rate of 1000 Hz. ECG data were then transferred to the MEG system.

Next, R-R wave variability was assessed. R-peak extraction and ectopic beat correction were performed using MemCalc for Windows (Global Medical Solution Inc., Tokyo, Japan). For frequency domain analysis of the R-R wave intervals, the R-R intervals were analyzed using the maximum entropy method with MemCalc for Windows. Low-frequency (LF) power was calculated as the power within the frequency range of 0.04–0.15 Hz, and high-frequency (HF) power was calculated as that within the frequency range of 0.15–0.4 Hz. LF and HF power were measured in absolute units ($ms^2$). The LF/HF ratio is considered to reflect sympathetic nervous system activity [39], and is thought to include important information about physiological processes because LH/HF ratio has been reported to be related to pathophysiological states such as mental stress and fatigue [40–44].

## Statistical analyses

All values are shown as mean ± SD unless otherwise stated. Two-way repeated-measures analysis of variance (ANOVA) was performed to assess alterations in the LF/HF ratio caused by viewing the food pictures compared with viewing the object pictures in the visible and invisible conditions. The LF/HF ratios were compared between the food and object sessions under the invisible condition using a paired $t$-test. Pearson's correlation analysis was performed to assess the associations between the interaction of neural activity and the eating behavior indices obtained from the questionnaires. Values of $P < 0.05$ were considered statistically significant. The $P$ values for the paired $t$-test and correlation analysis were two-tailed. All statistical analyses described above were performed using the IBM SPSS 21.0 software package (IBM, Armonk, NY). As described in the Results, the MEG data from 17 participants were excluded from the analysis because of the contamination of the MEG data by the magnetic noise originating from outside the shielded room, the insufficient backward masking effect, and so on.

## Results

### ECG analysis

Alterations in the LF/HF ratio were analyzed for 12 participants, as the MEG data from 17 participants were excluded from the analysis, as described below, and the ECG data from another two participants whose MEG data were analyzed were not obtained because of a technical error. Two-way repeated-measures ANOVA was performed to assess alterations in the LF/HF ratio caused by viewing the food pictures compared with viewing the object pictures in the visible and invisible conditions. A main effect of pictures (i.e., food or object pictures) was identified [$F(1, 11) = 21.637$, $P = 0.001$]. There was no main effect of conditions (i.e., visible or invisible conditions) [$F(1, 11) = 0.431$, $P = 0.525$] and no interaction was apparent [$F(1, 11) = 0.984$, $P = 0.343$]. Under the invisible condition, the LF/HF ratio in the food session was higher than that assessed in the object session ($P = 0.015$, paired $t$-test; Fig 2A).

### Spatial filtering analyses of the MEG data

The MEG data from 17 participants were excluded from the analysis: 13 participants were excluded from the analysis because 12 reported that they had recognized at least one food or object picture under the invisible condition and another reported that he could not recognize at least one food and/or object picture under the visible condition. Among the 12 participants who were dropped for stimulus recognition in the invisible condition, eight participants performed a visible > invisible sequence and four participants performed an invisible > visible sequence. The MEG data from the other four participants were contaminated with magnetic noise originating from outside the shielded room, and thus, the numbers of epochs remaining after excluding the contaminated ones were insufficient for the analysis (MEG data with > 24 epochs were analyzed). As a result, the MEG data from 14 participants were analyzed.

The neural activity corresponding to the interaction between sessions (i.e., food or object) and conditions (i.e., visible or invisible) was examined. Neural activity corresponding to an increase of high-gamma (80–200 Hz) oscillatory brain activity caused by viewing the pictures was found in left BA 45 and 47 ($P = 0.0029$ and $P = 0.0030$, respectively; family-wise error-

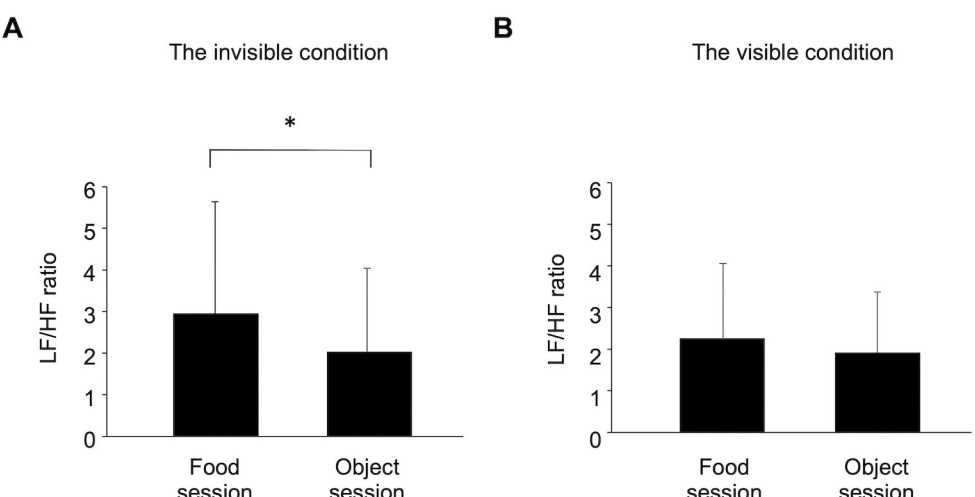

**Fig 2. Index of sympathetic nervous activity over each session.** (A) The LF/HF ratio assessed by frequency domain analysis of the R-R wave intervals in electrocardiograms (ECGs) under the invisible condition. The LF/HF ratio in the food session was higher than that in the object session. (B) The LF/HF ratio under the visible condition. *$P < 0.05$ (paired $t$-test). LF, low-frequency power; HF, high-frequency power.

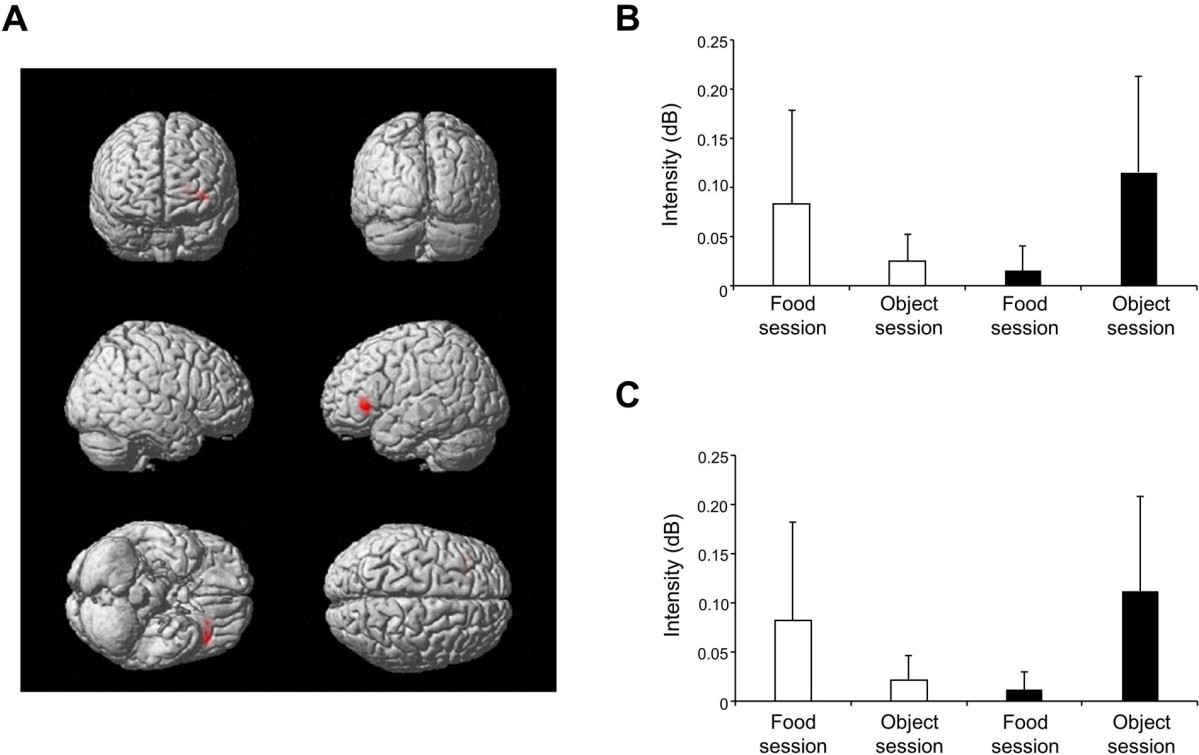

**Fig 3. Neural activity corresponding to the interaction between sessions and conditions.** (A) Statistical parametric map of the interaction observed between sessions (i.e., food or object) and conditions (i.e., visible or invisible) in the neural activity corresponding to the increase of high-gamma (80–200 Hz) oscillatory brain activity caused by viewing the pictures. The cluster in the left frontal region including Brodmann's areas (BA) 45 and 47 was above the statistical threshold set at $P = 0.00417$ (family-wise error-corrected for multiple comparisons). (B) The levels of increases in high-gamma oscillatory brain activity observed in each session in left BA 45. Open and closed columns represent the values under the visible and invisible conditions, respectively. (C) The levels of increases in high-gamma oscillatory brain activity observed in each session in left BA 47. Open and closed columns represent the values under the visible and invisible conditions, respectively.

corrected for multiple comparisons; Fig 3A and Table 1). The intensities of the high-gamma oscillatory brain activity observed in the food and object sessions under the visible and invisible conditions in BA 45 and 47 corresponding to the interaction are shown in Fig 3B and 3C, respectively.

Negative correlations were observed between the extent to which the neural responses to the pictures in the visible condition was different from those in the invisible condition in BA 45 and the index of emotional eating (r = −0.652, $P = 0.012$; Fig 4A), and between the extent to

**Table 1. Brain regions in which an interaction was observed between sessions and conditions.**

| Location | BA | MNI coordinate (mm) | | | Z value |
|---|---|---|---|---|---|
| | | x | y | z | |
| Inferior frontal gyrus | 45 | -28 | 28 | 5 | 4.19 |
| Inferior frontal gyrus | 47 | -38 | 28 | 0 | 4.18 |

BA, Brodmann area; MNI, Montreal Neurological Institute.

x, y, z: Stereotaxic coordinate.

Data were obtained from random-effect analysis. Only the results of the brain locations in which statistically significant changes were observed were shown (paired *t*-test, $P < 0.00417$, corrected for multiple voxel-wise comparisons with family-wise error rate).

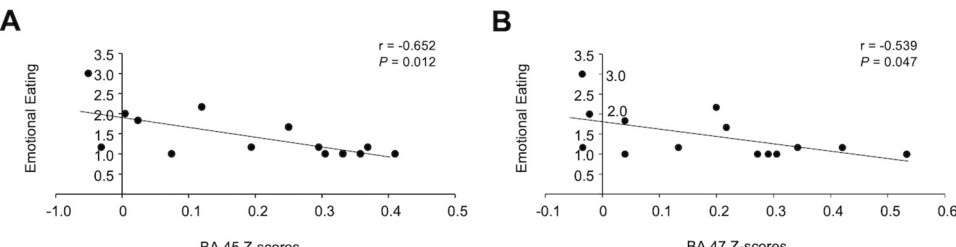

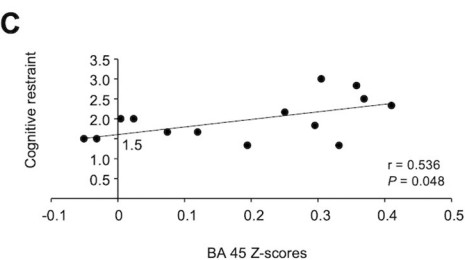

**Fig 4. Relationships between the indices of eating behavior and the extent to which the neural responses to the pictures in the visible condition was different from those in the invisible condition observed in BA 45 and 47.** (A) Relationship between the level of emotional eating and the extent to which the neural responses to the pictures in the visible condition was different from those in the invisible condition observed in BA 45. (B) Relationship between the level of emotional eating and the extent to which the neural responses to the pictures in the visible condition was different from those in the invisible condition observed in BA 47. (C) Relationship between the level of cognitive restraint and the interaction observed in BA 45. The linear regression line, Pearson's correlation coeffect, and *P* value are shown. The standardized values for the indices of emotional eating and cognitive restraint were used.

which the neural responses to the pictures in the visible condition was different from those in the invisible condition in BA 47 and the index of emotional eating (r = −0.539, *P* = 0.047; Fig 4B). A positive correlation was observed between the extent to which the neural responses to the pictures in the visible condition was different from those in the invisible condition in BA 45 and the index of cognitive restraint (r = 0.536, *P* = 0.048; Fig 4C).

## Discussion

In this study, participants viewed food and object pictures under visible and invisible conditions, respectively, and the neural activities over viewing the pictures were assessed by MEG. Under the invisible condition, the index of sympathetic nervous activity (i.e., the LF/HF ratio) in the food session was higher than that in the object session. The interaction between sessions (i.e., food or object) and conditions (i.e., visible or invisible) in the neural activity corresponding to the increase of high-gamma (80–200 Hz) oscillatory brain activity caused by viewing the pictures was observed in left BA 45 and 47. In addition, the extent to which the neural responses to the pictures in the visible condition was different from those in the invisible condition observed in BA 45 and 47 were negatively associated with the level of emotional eating, and that observed in BA 45 was positively associated with the level of cognitive restraint.

The data from individuals who answered that there was at least one picture in which they could not recognize a food item or object under the visible condition, and those from individuals who answered that there was at least one picture in which they recognized a food item or an object under the invisible condition were excluded from our analyses. Therefore, the participants whose data were analyzed reported that they have recognized food and object items

under the visible condition and have not recognize food and object items under the invisible condition.

The LF/HF ratio assessed in the food session was increased compared with that in the object session under the invisible condition. The LF/HF ratio has been reported to be increased after compared with before viewing food pictures presented below the threshold of awareness, which suggests that visual food stimuli affect the central nervous system even if presented below the threshold of awareness [18]; however, since mosaic pictures created from original food stimuli presented below the threshold of awareness were used as control stimuli in their study, the effects caused by the difference between objects (i.e., food items in this case) and non-objects (i.e., the mosaic pictures) and those caused by the difference between food and non-food objects could have both been included in the previous study. By using the pictures of objects (i.e., not the mosaic pictures) as control stimuli in our present study, we could confirm that the increase of the sympathetic nervous activity caused by viewing the food pictures presented using the backward masking procedure reported in the previous study was not due to the difference between objects and non-objects.

We hypothesized that if food pictures presented below the threshold of awareness play important roles such that affecting appetitive control and/or eating behavior, food pictures presented below the threshold of awareness may cause alterations of autonomic activity. In fact, it has been reported that, in a study in which affective pictures presented both below and above the threshold of awareness, alteration in heart rate variability was induced by both presentation conditions [45]. As a result, we showed that the LF/HF ratio was increased by viewing food pictures presented below the threshold of awareness in our previous study [18] and confirmed this finding in our present study, suggesting that the food pictures can affect neural activity even if presented below the threshold of awareness. Since it has been reported that food stimuli presented above the threshold of awareness caused increased low frequency (LF) power of heart rate variability in healthy participants [46] and that individuals with obesity showed altered vagal reactivity to food stimuli presented above the threshold of awareness compared with that of healthy individuals [47, 48], there may be a possibility that the alterations in heart rate variability caused by food pictures presented below the threshold of awareness are related to the disorders related to appetitive control including obesity; however, further studies are needed on this point.

There was no difference in the alteration of the sympathetic nervous activity caused by viewing the food pictures between the invisible and visible conditions (i.e., the main effect of conditions was not identified) in our present study, suggesting that the sympathetic nervous activity was increased by viewing the food pictures both in the visible and invisible conditions; however, since the participants whose data were analyzed reported that they did not recognize the food and object pictures exclusively in the invisible condition and the neural response to the visual food stimuli in the invisible condition was different from that in the visible condition as described below, it is speculated that the neural mechanisms by which the alteration of the sympathetic nervous activity was caused by viewing the food pictures in the invisible condition were not the same as those in the visible condition. Although we are not able to exclude the possibility that the visual food stimuli of which individuals reported not to be recognized worked as the sufficient stimuli to induce the alteration of the sympathetic nervous activity in the invisible condition, since there was no evidence that some of the features of the food stimuli presented in the invisible condition reached consciousness of our participants whose data were analyzed, it may be that the alteration of the sympathetic nervous activity caused in the visible condition was related to the unconscious processing of the food stimuli in the visible condition rather than that the alteration of the sympathetic nervous activity caused in the

invisible condition was due to the existence of partial awareness of the food stimuli in the invisible condition [49], if this is the case.

Interactions between sessions (i.e., food or object) and conditions (i.e., visible or invisible) were observed in left BA 45 and 47 in the high-gamma frequency range, suggesting that the food pictures presented below the threshold of awareness acted on the brain regions in different way than did those presented above the threshold of awareness (Fig 3B and 3C). It is of note that the visual presentation in the visible condition was comparable to that in the invisible condition except for the difference of the vertical position of the mask picture on the screen and that the variable we focused on was the interaction in which the effects caused across the visibility was canceled out between the food and object conditions. Some studies have reported that oscillatory brain activity in the high-gamma frequency range is related to cognitive processes such as attention and conscious perceptions [25–28]. In addition, coordinated gamma (30–90 Hz) oscillations in the prefrontal cortex have been reported to be necessary for food-seeking behavior [50]. Furthermore, the inferior frontal gyrus seems to be involved in inhibitory control [51–54] and that of appetite [18]. Taking these findings into consideration, it is speculated that the interaction in high-gamma oscillatory brain activity observed in BA 45 and 47 is involved in cognitive processes related to eating behavior.

In fact, the extent to which the neural responses to the pictures in the visible condition was different from those in the invisible condition observed in this study was associated with the indices of eating behavior. The extent to which the neural responses to the pictures in the visible condition was different from those in the invisible condition observed in BA 45 and 47 were negatively associated with the level of emotional eating, and that observed in BA 45 was positively associated with the level of cognitive restraint index, confirming the speculation that the interactions in high-gamma oscillatory brain activity observed in BA 45 and 47 were involved in cognitive processes related to the indices of eating behavior. These findings indicate that it is essential to taking both the neural responses to visual food stimuli presented above the threshold of awareness and those below the threshold of awareness into consideration to understand the neural mechanisms related to eating behavior, although we could not determine the causal association between the existence of the interaction in the oscillatory brain activity and the specific features of eating behavior (i.e., emotional eating and cognitive restraint) in our present study.

The interactions were observed only in high-gamma band frequency. This suggests the importance of high-gamma band neural activity in investigating the conscious and unconscious neural processes related to eating behavior; however, further studies were need to completely understand the differential roles of neural activities in eating behavior between high-gamma band and gamma bands 1 and 2.

This study did have some limitations. First, the participants were all healthy males; therefore, additional studies involving females and/or obese individuals need to be conducted to gain a better understanding of the role of the interaction with eating behavior. Second, in this study, we focused on the cortical neural activity in the frontal brain regions because we were interested in the neural activity related to the regulation of eating behavior and/or appetite rather than that specifically related to the modality of the stimuli (i.e., the neural activity in the visual cortex in this case), and because the sensitivity of MEG for source activity in deep brain regions (i.e., subcortical brain regions) is low compared with that in cortical brain regions. In future research, it would be beneficial to investigate the neural mechanisms of conscious and unconscious neural processes related to eating behavior in terms of network dynamics within the brain, including the cortical and subcortical regions, which were not examined in this study. Third, since another arousing stimulus other than food images was not employed as a control stimulus in our present study, we are not able to exclude the possibility that the effects

caused by viewing the food images might have been due to arousal induced by the images in our present study; however, we think that it is valuable to assess the neural effects caused by food images including those related to arousal induced by the images when we assess the neural activity caused by viewing food images because arousal can inevitably be induced by viewing food and/or food images in our daily lives. Fourth, since we wanted to ensure that, in the visible condition, the food and object pictures presented for the same duration to that in the invisible condition could be recognized by most of the participants and thus the duration for the presentation of the food and object pictures was longer than that would have been able to make the pictures invisible for all the participants in the invisible condition, only the MEG data for the participants who recognized no food or object pictures under the invisible condition (and recognized all the food and object pictures under the visible condition) were analyzed. Since the neural mechanisms by which individual differences in eating behavior affect the visibility of the masked food pictures are unknown, further study is needed on this point. In addition, since longer experimental time period is not optimum for MEG data collection from the perspective of minimizing the body movement during the data collection and the burden of participants, we decided to ask them whether they recognized images other than the mask image after each session. To minimize the possibility that the MEG data of the participants who in fact recognized some of the food or object items in the invisible condition were included in our analysis, we excluded the MEG data of the participants who declared that they might have seen items other than the mask picture at least one time in the invisible condition regardless of whether they were able to mention what they saw or whether the items they mentioned were actually presented in the session. Fifth, the vertical position of the mask image in the visible condition was not the same as that in the invisible condition because we wanted the time course of the presentation in the visible condition to be the same as that in the invisible condition; however, since we assessed the alterations in neural responses corresponding to [(food pictures in visible condition + object pictures in invisible condition)–(object pictures in visible condition + food pictures in invisible condition)] in our present study, the effect caused by the mask image was canceled out in the analysis (i.e., the contrast described above was equal to [(food pictures in visible condition–object pictures in visible condition)–(food pictures in invisible condition–object pictures in invisible condition)] and therefore, the effect caused by the difference of the position of the mask picture between the conditions seems to be minimal in our present study. Likewise, although the items presented in the invisible condition were the same as that presented in the visible condition, since the contrast corresponding to [(food pictures in visible condition–object pictures in visible condition)–(food pictures in invisible condition–object pictures in invisible condition] was analyzed in our present study, the memory effect was expected to be canceled out in the process of the calculation of the contrast (i.e., the possible effects caused by the visible > invisible and/or the invisible > visible sequences existed both in the food and object sessions).

In summary, we confirmed that increased sympathetic nervous activity is induced by viewing food pictures presented below the threshold of awareness, as reported previously [18], and that conscious and unconscious neural processes involved differently in the neural processes related to eating behavior. Although our present study is in the proof of principle stage only to show the potential importance of unconscious neural processes in eating behavior, together with the reports that unconscious decision-making processes have effects on appetitive behavior mentioned in the Introduction [15, 16], our findings can be expected to motivate further studies seeking to clarify the neural mechanisms related to eating behavior from the perspectives of conscious and unconscious neural processes, and to provide valuable clues to develop more effective methods for assessing conscious and unconscious regulations of appetite in individuals with normal and abnormal eating behaviors.

## Acknowledgments

We wish to thank Forte Science Communications for editorial help with the manuscript and Manryoukai Imaging Clinic for assistance with the MRI scans.

## Author Contributions

**Conceptualization:** Rika Ishida, Akira Ishii.

**Data curation:** Rika Ishida, Akira Ishii, Takashi Matsuo.

**Formal analysis:** Rika Ishida, Akira Ishii, Takashi Matsuo.

**Funding acquisition:** Akira Ishii.

**Investigation:** Rika Ishida, Akira Ishii, Takashi Matsuo, Takayuki Minami.

**Methodology:** Rika Ishida, Akira Ishii, Takashi Matsuo.

**Project administration:** Akira Ishii.

**Software:** Akira Ishii.

**Supervision:** Akira Ishii, Takahiro Yoshikawa.

**Validation:** Akira Ishii.

**Visualization:** Rika Ishida, Akira Ishii.

**Writing – original draft:** Rika Ishida, Akira Ishii.

**Writing – review & editing:** Akira Ishii.

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
