## [Decision Letter · Decision Letter 0]

27 Mar 2021

PONE-D-20-30494

Association between eating behavior and the immediate neural activity caused by viewing food images presented in and out of awareness: a magnetoencephalography study

PLOS ONE

Dear Dr. Ishii,

Thank you for submitting your manuscript to PLOS ONE. After careful consideration, we feel that it has merit but does not fully meet PLOS ONE’s publication criteria as it currently stands. Therefore, we invite you to submit a revised version of the manuscript that addresses the points raised during the review process.

The reviewer has pointed out a number of issues to be rectfied our clarified.

We look forward to receiving your revised manuscript.

Kind regards,

Pedro Antonio Valdes-Sosa, Ph.D., M.D.

Academic Editor

PLOS ONE

Journal Requirements:

Reviewers' comments:

Reviewer's Responses to Questions

**Comments to the Author**

1. Is the manuscript technically sound, and do the data support the conclusions?

Reviewer #1: Partly

2. Has the statistical analysis been performed appropriately and rigorously? 

Reviewer #1: I Don't Know

3. Have the authors made all data underlying the findings in their manuscript fully available?

Reviewer #1: Yes

4. Is the manuscript presented in an intelligible fashion and written in standard English?

Reviewer #1: Yes

5. Review Comments to the Author

Reviewer #1: This study examined whether briefly food stimuli presented below the level of conscious awareness influences sympathetic arousal and brain activity and whether these responses differed from the same stimuli when consciously recognized and from non-food objects. The current study builds nicely on the authors’ previously-published pilot study. On one hand, the question of whether nonconsciously-perceived food stimuli affect physiological response – and, perhaps, eating behavior – is an important one. Therefore, the authors’ demonstration of some differential responses to unconsciously and consciously presented food stimuli is of potential interest. On the other hand, there were a large number of serious theoretical, methodological and interpretive problems with the study that need to be addressed.

The statement on lines 54-58 is an intriguing possibility but only one reference was given in support of it and that reference does not provide any support for the notion that unconscious perception of food stimuli helps explain the poor outcomes of behavioral weight loss programs. The authors’ hypotheses about unconscious reactions to food need to be recognized as highly speculative at this point and not try to oversell (here and elsewhere in the manuscript) that these effects are even real, never mind relevant to weight control. This literature is still very much in the “proof of principle” stage and the authors’ narrative throughout their paper should reflect this.

A major issue throughout the manuscript is that the authors do not realize that food has both specific and non-specific effects on physiology. As a highly rewarding substance, food creates general arousal (and, for dieters, may create an approach-avoidance conflict) above and beyond its specific appetitive effects. So in their discussion of the literature and also in their study, they should discuss that fact that general arousal could account for some or all of the effects they are currently attributing to food. Ideally the authors would have utilized a second control condition, using another type of arousing stimulus like sexually arousing pictures. This limitation needs to be recognized throughout the paper.

In several places in the paper, the authors make statements about their own or others’ research findings without specifying whether the results were based on unconscious or conscious stimuli (e.g., lines 90-93). Please revise accordingly.

The authors often refer to the regulation of eating or the measurement of food intake, but they are simply referring to scores on self-report measures like emotional eating. The validity of these questionnaires is questionable and so the authors should instead be referring, for example, to “scores on measures of emotional eating,” not to “eating behavior” or “eating regulation.”

The authors need to explain why they didn’t present visible and invisible stimuli in the same session (e.g., lines 117-119).

Lines 144-146 and elsewhere: what were the “mask pictures” of?

No criteria were given for why certain Japanese foods were used as stimuli. How were they chosen? Were they known to represent highly palatable foods?

A major problem with the study is that it included a measure of heart rate variability and of brain electrical activity but there was little explanation of why these two physiological measures were chosen or how they might relate to one another. They chose a measure of heart rate variability and, though there is a large literature on the relation between HRV and appetitive stimuli, almost none of this was included. Further, please explain what is thought to reflect healthy or unhealthy heart rate variability and why this might be related to perception of food stimuli.

On lines 263-267, the authors should analyze these data with repeated measures ANOVAs, not separate t-tests.

I have never heard of analyzing the “intensity of an interaction.” The reason an interaction is interesting is because it captures different responses on a dependent variable depending on the level of an independent variable. One would presumably want to separately capture both patterns that comprise an interaction. What does intensity of an interaction even mean and is there precedent in the literature for using it? Also, I have no idea what the relationships in Figure 4 refer to because the “strength of the interaction” has no apparent meaning.

There was no mention of counterbalancing stimuli. Also, if conscious food stimuli were presented first, couldn’t that suggest to participants what the invisible stimuli were?

The fact that a large fraction of the participants claimed to have seen the “unconscious” food stimuli and therefore had to be eliminated is a major methodological flaw that the authors do not seem to recognize. This suggests that they did not pilot test their subliminal presentation of stimuli to see if they “worked.” Those who did and did not recognize the food stimuli should be compared on all measures to see if they differed in any way (using effect sizes rather than significance levels, since the sample sizes are so small).

The authors discuss findings in high-gamma range at length but do not mention why no findings existed for low- or medium-level gamma. What might this signify?

I do not understand the potential meaning of the sharply different patterns of gamma in Figures 3 A and B. It seems to me that what we have here are findings that transcend any appetitive model or theory of why this pattern may exist. In other words, just because our technical capabilities can detect differential brain responding doesn’t mean that the field has sufficient knowledge to interpret what these responses might mean in terms of cognition or behavior. Unless the authors have a better speculation about the meaning of their findings than what they currently provide, this may be the best conclusion they can reach at present.

6. PLOS authors have the option to publish the peer review history of their article (what does this mean?). If published, this will include your full peer review and any attached files.

Reviewer #1: No

---

## [Author Response · Author response to Decision Letter 0]

13 May 2021

Responses to the comments

Reviewer 1

1. The statement on lines 54-58 is an intriguing possibility but only one reference was given in support of it and that reference does not provide any support for the notion that unconscious perception of food stimuli helps explain the poor outcomes of behavioral weight loss programs. The authors’ hypotheses about unconscious reactions to food need to be recognized as highly speculative at this point and not try to oversell (here and elsewhere in the manuscript) that these effects are even real, never mind relevant to weight control. This literature is still very much in the “proof of principle” stage and the authors’ narrative throughout their paper should reflect this.

Response:

Thank you for the suggestion. We added the descriptions in the Discussion to clearly state that the idea that appetitive behavior is driven more by unconscious decision-making processes than by the conscious decision-making processes is speculative at this point: “the idea that appetitive behavior is driven more by unconscious decision-making processes than by the conscious ones is speculative at this point and our present study is in the proof of principle stage only to show the potential importance of unconscious neural processes in eating behavior” (lines 486 to 489, in the revised manuscript).

2. A major issue throughout the manuscript is that the authors do not realize that food has both specific and non-specific effects on physiology. As a highly rewarding substance, food creates general arousal (and, for dieters, may create an approach-avoidance conflict) above and beyond its specific appetitive effects. So in their discussion of the literature and also in their study, they should discuss that fact that general arousal could account for some or all of the effects they are currently attributing to food. Ideally the authors would have utilized a second control condition, using another type of arousing stimulus like sexually arousing pictures. This limitation needs to be recognized throughout the paper.

Response:

We agree with the Reviewer 1 that we are not able to exclude the possibility that the effects caused by viewing the food images might have been due to arousal induced by the images in our present study; however, we think that it is of value to assess the neural effects caused by food images including those related to arousal induced by the images when we assess the neural activity caused by viewing food images because arousal can inevitably be induced by viewing food and/or food images in our daily lives. We added the descriptions, “It is of note that the effects caused by viewing the food images may include those of arousal induced by the images (i.e., another arousing stimulus other than food images was not employed as a control stimulus in our present study)” (lines 103 to 105, in the revised manuscript) and “Third, since another arousing stimulus other than food images was not employed as a control stimulus in our present study, we are not able to exclude the possibility that the effects caused by viewing the food images might have been due to arousal induced by the images in our present study; however, we think that it is valuable to assess the neural effects caused by food images including those related to arousal induced by the images when we assess the neural activity caused by viewing food images because arousal can inevitably be induced by viewing food and/or food images in our daily lives” (lines 465 to 472, in the revised manuscript).

3. In several places in the paper, the authors make statements about their own or others’ research findings without specifying whether the results were based on unconscious or conscious stimuli (e.g., lines 90-93). Please revise accordingly.

Response:

Thank you for the suggestion. We revised our manuscript as suggested by the Reviewer 1 (lines 90 to 91 and lines 92 to 93, in the revised manuscript).

4. The authors often refer to the regulation of eating or the measurement of food intake, but they are simply referring to scores on self-report measures like emotional eating. The validity of these questionnaires is questionable and so the authors should instead be referring, for example, to “scores on measures of emotional eating,” not to “eating behavior” or “eating regulation.”

Response:

We revised our manuscript as suggested by the Reviewer 1 (lines 83, 84, 354, 356, 359, 381 to 382, 437, 440, 442, 444, 446, and 448, in the revised manuscript).

5. The authors need to explain why they didn’t present visible and invisible stimuli in the same session (e.g., lines 117-119).

Response:

We wanted to confirm that our participants were not aware of the food and object images in the invisible trials and were aware of the food and object images in the visible trials by asking them whether they recognized images other than the mask image afterward and therefore, if the visible and invisible stimuli were randomly presented in the same session, we had to ask our participants whether they recognized images other than the mask image after each trial, resulting in much longer experimental time period for each participant, which is not optimum for MEG data collection. We added the description, “Since we wanted to confirm that our participants were not aware of the food and object images in the invisible trials and were aware of the food and object images in the visible trials by asking them whether they recognized images other than the mask image afterward, as described below, the visible and invisible stimuli were not presented in the same session” (lines 122 to 126, in the revised manuscript).

6. Lines 144-146 and elsewhere: what were the “mask pictures” of?

Response:

The mask picture was the same for every trial across the food and object sessions in the visible and invisible conditions: The mask picture was created by arranging colored lines and circles and did not created by processing specific picture. We added the sentence, “The mask picture was the same for every trial across the food and object sessions in the visible and invisible conditions: The mask picture was created by arranging colored lines and circles and did not created by processing specific picture” (lines 155 to 157, in the revised manuscript), in the Materials and methods.

7. No criteria were given for why certain Japanese foods were used as stimuli. How were they chosen? Were they known to represent highly palatable foods?

Response:

The set of food pictures used in our present study was identical to those used in our previous studies (Yoshikawa et al., 2013, 2014a, 2014b; Takada et al., 2018; Nakamura et al., 2020). We chose food items which are thought to be consumed in daily living in Japan, including highly palatable food such as steak, hamburger, pizza, fried rice, and takoyaki (octopus balls). We revised the description in the Materials and methods (lines 158 to 161, in the revised manuscript)

References: 

1. Yoshikawa T, Tanaka M, Ishii A, Watanabe Y. Immediate neural responses of appetitive motives and its relationship with hedonic appetite and body weight as revealed by magnetoencephalography. Medical science monitor : international medical journal of experimental and clinical research. 2013;19:631-40. Epub 2013/08/03. doi: 10.12659/MSM.889234. PubMed PMID: 23907366; PubMed Central PMCID: PMC3737122.

2. Yoshikawa T, Tanaka M, Ishii A, Watanabe Y. Suppressive responses by visual food cues in postprandial activities of insular cortex as revealed by magnetoencephalography. Brain Res. 2014;1568:31-41. Epub 2014/04/29. doi: 10.1016/j.brainres.2014.04.021. PubMed PMID: 24768717.

3. Yoshikawa T, Tanaka M, Ishii A, Fujimoto S, Watanabe Y. Neural regulatory mechanism of desire for food: revealed by magnetoencephalography. Brain Res. 2014;1543:120-7. Epub 2013/11/13. doi: 10.1016/j.brainres.2013.11.005. PubMed PMID: 24216133.

4. Takada K, Ishii A, Matsuo T, Nakamura C, Uji M, Yoshikawa T. Neural activity induced by visual food stimuli presented out of awareness: a preliminary magnetoencephalography study. Scientific reports. 2018;8(1):3119. Epub 2018/02/17. doi: 10.1038/s41598-018-21383-0. PubMed PMID: 29449657; PubMed Central PMCID: PMC5814400.

5. Nakamura C, Ishii A, Matsuo T, Ishida R, Yamaguchi T, Takada K, et al. Neural effects of acute stress on appetite: A magnetoencephalography study. PLoS One. 2020;15(1):e0228039. Epub 2020/01/23. doi: 10.1371/journal.pone.0228039. PubMed PMID: 31968008; PubMed Central PMCID: PMCPMC6975544.

8. A major problem with the study is that it included a measure of heart rate variability and of brain electrical activity but there was little explanation of why these two physiological measures were chosen or how they might relate to one another. They chose a measure of heart rate variability and, though there is a large literature on the relation between HRV and appetitive stimuli, almost none of this was included. Further, please explain what is thought to reflect healthy or unhealthy heart rate variability and why this might be related to perception of food stimuli.

Response:

In our previous study (Takada et al., 2018), we hypothesized that if food pictures presented below the threshold of awareness play important roles such that affecting appetitive control and/or eating behavior, food pictures presented below the threshold of awareness may cause alterations of autonomic activity. In fact, it has been reported that, in a study in which affective pictures presented both below and above the threshold of awareness, alteration in heart rate variability was induced by both presentation conditions (Bulut et al., 2018). As a result, we showed that the LF/HF ratio was increased by viewing food pictures presented below the threshold of awareness in our previous study (Takada et al., 2018) and confirmed this finding in our present study, suggesting that the food pictures can affect neural activity even if presented below the threshold of awareness. As the Reviewer 1 suggested, there have been reports on the relation between HRV and appetitive stimuli. For example, it has been reported that food stimuli presented above the threshold of awareness caused increased low frequency (LF) power of heart rate variability in healthy participants (Nederkoorn et al., 2000) and that individuals with obesity showed altered vagal reactivity to food stimuli presented above the threshold of awareness compared with that of healthy individuals (Udo et al., 2014 and Spitoni et al., 2017). Therefore, there may be a possibility that the alterations in heart rate variability caused by food pictures presented below the threshold of awareness are related to the disorders related to appetitive control including obesity; however, further studies are needed on this point. We added the descriptions, “We hypothesized that if food pictures presented below the threshold of awareness play important roles such that affecting appetitive control and/or eating behavior, food pictures presented below the threshold of awareness may cause alterations of autonomic activity. In fact, it has been reported that, in a study in which affective pictures presented both below and above the threshold of awareness, alteration in heart rate variability was induced by both presentation conditions (Bulut et al., 2018). As a result, we showed that the LF/HF ratio was increased by viewing food pictures presented below the threshold of awareness in our previous study (Takada et al., 2018) and confirmed this finding in our present study, suggesting that the food pictures can affect neural activity even if presented below the threshold of awareness. Since it has been reported that food stimuli presented above the threshold of awareness caused increased low frequency (LF) power of heart rate variability in healthy participants (Nederkoorn et al., 2000) and that individuals with obesity showed altered vagal reactivity to food stimuli presented above the threshold of awareness compared with that of healthy individuals (Udo et al., 2014 and Spitoni et al., 2017), there may be a possibility that the alterations in heart rate variability caused by food pictures presented below the threshold of awareness are related to the disorders related to appetitive control including obesity; however, further studies are needed on this point”, in the Discussion (lines 404 to 419, in the revised manuscript).

References:

1. Takada K, Ishii A, Matsuo T, Nakamura C, Uji M, Yoshikawa T. Neural activity induced by visual food stimuli presented out of awareness: a preliminary magnetoencephalography study. Scientific reports. 2018;8(1):3119. Epub 2018/02/17. doi: 10.1038/s41598-018-21383-0. PubMed PMID: 29449657; PubMed Central PMCID: PMC5814400.

2. Bulut NS, Wurz A, Yorguner Kupeli N, Carkaxhiu Bulut G, Sungur MZ. Heart rate variability response to affective pictures processed in and outside of conscious awareness: Three consecutive studies on emotional regulation. Int J Psychophysiol. 2018;129:18-30. Epub 2018/05/23. doi: 10.1016/j.ijpsycho.2018.05.006. PubMed PMID: 29787784.

3. Nederkoorn C, Smulders FT, Jansen A. Cephalic phase responses, craving and food intake in normal subjects. Appetite. 2000;35(1):45-55. Epub 2000/07/18. doi: 10.1006/appe.2000.0328. PubMed PMID: 10896760.

4. Udo T, Weinberger AH, Grilo CM, Brownell KD, DiLeone RJ, Lampert R, et al. Heightened vagal activity during high-calorie food presentation in obese compared with non-obese individuals--results of a pilot study. Obes Res Clin Pract. 2014;8(3):e201-98. Epub 2014/05/23. doi: 10.1016/j.orcp.2013.05.006. PubMed PMID: 24847667; PubMed Central PMCID: PMCPMC4031442.

5. Spitoni GF, Ottaviani C, Petta AM, Zingaretti P, Aragona M, Sarnicola A, et al. Obesity is associated with lack of inhibitory control and impaired heart rate variability reactivity and recovery in response to food stimuli. Int J Psychophysiol. 2017;116:77-84. Epub 2017/04/10. doi: 10.1016/j.ijpsycho.2017.04.001. PubMed PMID: 28390903.

9. On lines 263-267, the authors should analyze these data with repeated measures ANOVAs, not separate t-tests.

Response:

According to the Reviewer 1’s suggestion, we added the results of repeated measures ANOVA. We added the description, “Two-way repeated-measures analysis of variance (ANOVA) was performed to assess alterations in the LF/HF ratio caused by viewing the food pictures compared with viewing the object pictures in the visible and invisible conditions” (lines 268 to 271, in the revised manuscript), in the Materials and methods and revised the descriptions in the Results as follows: “Two-way repeated-measures ANOVA was performed to assess alterations in the LF/HF ratio caused by viewing the food pictures compared with viewing the object pictures in the visible and invisible conditions. A main effect of pictures (i.e., food or object pictures) was identified [F(1, 11) = 21.637, P = 0.001]. There was no main effect of conditions (i.e., visible or invisible conditions) [F(1, 11) = 0.431, P = 0.525] and no interaction was apparent [F(1, 11) = 0.984, P = 0.343]” (lines 286 to 291, in the revised manuscript).

10. I have never heard of analyzing the “intensity of an interaction.” The reason an interaction is interesting is because it captures different responses on a dependent variable depending on the level of an independent variable. One would presumably want to separately capture both patterns that comprise an interaction. What does intensity of an interaction even mean and is there precedent in the literature for using it? Also, I have no idea what the relationships in Figure 4 refer to because the “strength of the interaction” has no apparent meaning.

Response:

Thank you for the suggestion. The descriptions regarding this point were inadequate in our previous manuscript: We showed the existence of the brain regions which responded differently to the food pictures between the visible and invisible conditions. As shown in Figure 3B and 3C, the neural activities caused by viewing the food and object pictures were sharply different between the visible and invisible conditions (please refer to the Response to the Reviewer 1’s comment #14). Since the extent to which the neural responses to the pictures in the visible condition was different from those in the invisible condition was variable between participants, we wanted to assess whether the extent to which the neural responses to the pictures in the visible condition was different from those in the invisible condition was related to the indices of eating behavior. We assessed the extent to which the neural responses to the pictures in the visible condition was different from those in the invisible condition by the values calculated as the contrast of the levels of oscillatory brain activity [(food pictures in visible condition + object pictures in invisible condition) – (object pictures in visible condition + food pictures in invisible condition)] for each participant, corresponding to the interactions assessed in the SPM analyses. We added the description, “The extent to which the neural responses to the pictures in the visible condition was different from those in the invisible condition was assessed by the values calculated as the contrast of the levels of oscillatory brain activity [(food pictures in visible condition + object pictures in invisible condition) – (object pictures in visible condition + food pictures in invisible condition)] for each participant, corresponding to the interactions assessed in the SPM analyses”, in the Materials and methods (lines 233 to 238, in the revised manuscript). We revised the expressions “the intensity of the interaction” and “the interaction” to be “the extent to which the neural responses to the pictures in the visible condition was different from those in the invisible condition” in our revised manuscript (lines 352 to 359, lines 364 to 370, lines 379 to 380, lines 435 to 437, lines 437 to 438, lines 446 to 447, and Figure 4).

11. There was no mention of counterbalancing stimuli. Also, if conscious food stimuli were presented first, couldn’t that suggest to participants what the invisible stimuli were?

Response:

As described in the Materials and methods (lines 126 to 127, in the revised manuscript) and shown in Figure 1, our experiment was performed in a four-crossover fashion (i.e., approximately half of the participants performed the visible condition before performing the invisible condition). We analyzed data from the participants who declared that they had not recognized even the existence of any pictures other than the mask picture in the invisible condition and the participants were not informed of the existence of pictures other than the mask picture in the invisible condition before the stat of the experiment. Therefore, all the participants whose data were analyzed seemed to have believed that the mask image and the fixation cross were the only images presented during the invisible condition.

12. The fact that a large fraction of the participants claimed to have seen the “unconscious” food stimuli and therefore had to be eliminated is a major methodological flaw that the authors do not seem to recognize. This suggests that they did not pilot test their subliminal presentation of stimuli to see if they “worked.” Those who did and did not recognize the food stimuli should be compared on all measures to see if they differed in any way (using effect sizes rather than significance levels, since the sample sizes are so small).

Response:

We intentionally employed longer duration for the presentation of the food and object pictures than our previous study (Takada et al., 2018), which led to the increased number of the participants who recognized the food and/or object pictures in the invisible condition, because we wanted to ensure that, in the visible condition, the food and object pictures presented for the same duration to that in the invisible condition could be recognized by most of the participants. In addition, since we planned to perform group analyses of the MEG data, we also wanted to use the same duration for the presentation of the food and object pictures for every participant. Therefore, the MEG data for the participants who recognized no food or object pictures under the invisible condition (and recognized all the food and object pictures under the visible condition) were analyzed in our present study.

According to the suggestion made by the Reviewer 1, we compared the participants whose MEG data were analyzed with those whose MEG data were not analyzed on the measures obtained in our present study (i.e., the indices related to eating behavior; Table 1). We added the descriptions, “The indices related to eating behavior assessed by using TFEQ (i.e., uncontrolled eating, cognitive restraint, and emotional eating) in the participants whose MEG data were analyzed and those whose MEG data were not analyzed were shown (Table 1)” and “The effect sizes comparing the participants whose MEG data were analyzed with those whose MEG data were not analyzed on the indices related to eating behavior were calculated by using G*Power software (Version 3. 1. 9. 2; Faul et al., 2007)”, in the Results (lines 313 to 316, in the revised manuscript) and Materials and methods (lines 278 to 281, in the revised manuscript), respectively. We also added the sentences in the Discussion as follows: “Since we wanted to ensure that, in the visible condition, the food and object pictures presented for the same duration to that in the invisible condition could be recognized by most of the participants and thus the duration for the presentation of the food and object pictures was longer than that would have been able to make the pictures invisible for most of the participants in the invisible condition, only the MEG data for the participants who recognized no food or object pictures under the invisible condition (and recognized all the food and object pictures under the visible condition) were analyzed. Although the relationship between the individual differences on whether the masked food pictures are recognized or not in the invisible condition and those of the indices of eating behavior is unknown, our results (Table 1) suggest the possibility that the differences in the indices of eating behavior have some influence on the visibility of the masked food pictures. Further study is needed on this point” (lines 472 to 482, in the revised manuscript).

References:

1. Takada K, Ishii A, Matsuo T, Nakamura C, Uji M, Yoshikawa T. Neural activity induced by visual food stimuli presented out of awareness: a preliminary magnetoencephalography study. Scientific reports. 2018;8(1):3119. Epub 2018/02/17. doi: 10.1038/s41598-018-21383-0. PubMed PMID: 29449657; PubMed Central PMCID: PMC5814400.

2. Faul F, Erdfelder E, Lang AG, Buchner A. G*Power 3: a flexible statistical power analysis program for the social, behavioral, and biomedical sciences. Behavior research methods. 2007;39(2):175-91. Epub 2007/08/19. PubMed PMID: 17695343.

13. The authors discuss findings in high-gamma range at length but do not mention why no findings existed for low- or medium-level gamma. What might this signify?

Response:

As the Reviewer 1 suggested, the findings were observed only in high-gamma band frequency. On the one hand, this suggests the importance of high-gamma band neural activity in investigating the conscious and unconscious neural processes related to eating behavior. On the other hand, we think it is premature to state the differential roles of neural activities in eating behavior between high-gamma and low- and medium-gamma frequency bands. We added the description, “The interactions were observed only in high-gamma band frequency. This suggests the importance of high-gamma band neural activity in investigating the conscious and unconscious neural processes related to eating behavior; however, further studies were needed to completely understand the differential roles of neural activities in eating behavior between high-gamma band and gamma bands 1 and 2”, in the Discussion (lines 449 to 453, in the revised manuscript). 

14. I do not understand the potential meaning of the sharply different patterns of gamma in Figures 3 A and B. It seems to me that what we have here are findings that transcend any appetitive model or theory of why this pattern may exist. In other words, just because our technical capabilities can detect differential brain responding doesn’t mean that the field has sufficient knowledge to interpret what these responses might mean in terms of cognition or behavior. Unless the authors have a better speculation about the meaning of their findings than what they currently provide, this may be the best conclusion they can reach at present.

Response:

We hypothesized that there may be an interaction between the conscious and unconscious neural process related to eating behavior if the unconscious neural process related to eating behavior has effects which are not explained by the conscious one and we showed the existence of the brain regions which responded differently to the food pictures between the visible and invisible conditions in our present study (Figure 3A and Table 2). The neural activity caused by viewing each visual stimulus (i.e., the food and object pictures in the visible and invisible conditions) in the each brain region which responded differently to the food pictures between the visible and invisible conditions was shown in Figure 3B and 3C: The neural activities caused by viewing the pictures were sharply different between the visible and invisible conditions as the Reviewer 1 suggested. Since the extents of the differences in the neural activity caused by viewing the pictures between the visible and invisible conditions were related to the indices of eating behavior, it can be thought that the differences in the neural activity caused by viewing the pictures seem to have some influences on eating behavior. However, as the Reviewer 1 suggested, we were not able to refer to the precise mechanisms by which the differences in the neural activities shown in Figure 3B and 3C affect eating behavior at this time. We revised the descriptions in the Discussion as follows: “we could not determine the causal association between the differences of the neural responses to the pictures between the visible and invisible conditions and the indices of eating behavior and the precise mechanisms by which the differences in the neural activities caused by viewing the food pictures between the visible and invisible conditions affect eating behavior if the differences in the neural activities affect eating behavior in this study” (lines 441 to 446, in the revised manuscript).

END

---

## [Decision Letter · Decision Letter 1]

17 May 2022

PONE-D-20-30494R1Association between eating behavior and the immediate neural activity caused by viewing food images presented in and out of awareness: a magnetoencephalography studyPLOS ONE

Dear Dr. Ishii,

Thank you for submitting your manuscript to PLOS ONE. After careful consideration, we feel that it has merit but does not fully meet PLOS ONE’s publication criteria as it currently stands. Therefore, we invite you to submit a revised version of the manuscript that addresses the points raised during the review process.

The reviewers felt that a number of concerns with the manuscript remain. Reviewer 3 in particular raised concerns with the study design, the presentation of methodological details and aspects of the statistical analysis. The reviewers' concerns can be viewed in full, below.

We look forward to receiving your revised manuscript.

Kind regards,

Natasha McDonald, PhD

Associate Editor

PLOS ONE

Reviewers' comments:

Reviewer's Responses to Questions

**Comments to the Author**

1. If the authors have adequately addressed your comments raised in a previous round of review and you feel that this manuscript is now acceptable for publication, you may indicate that here to bypass the “Comments to the Author” section, enter your conflict of interest statement in the “Confidential to Editor” section, and submit your "Accept" recommendation.

Reviewer #2: All comments have been addressed

Reviewer #3: All comments have been addressed

2. Is the manuscript technically sound, and do the data support the conclusions?

Reviewer #2: Yes

Reviewer #3: No

3. Has the statistical analysis been performed appropriately and rigorously? 

Reviewer #2: Yes

Reviewer #3: Yes

4. Have the authors made all data underlying the findings in their manuscript fully available?

Reviewer #2: Yes

Reviewer #3: No

5. Is the manuscript presented in an intelligible fashion and written in standard English?

Reviewer #2: Yes

Reviewer #3: Yes

6. Review Comments to the Author

Reviewer #2: This study was performed on healthy male volunteers who viewed pictures of food and non-food items presented both above and below the awareness threshold and the oscillatory brain activity affected by viewing the pictures was assessed by using MEG. The authors showed that neural activity corresponding to the interactions between sessions and conditions was observed in left Brodmann’s areas 45 and 47 in the high-gamma (60–200 Hz) frequency range. They demonstrated that conscious and unconscious neural processes are differently involved in eating behavior.

This paper is well written and the authors seems to adequately respond to the Reviewer 1’s comment, I think that this paper is acceptable for PLOS ONE.

However, there are some minor concerns before the acceptance.

Concerns

1. In Line 261, it is described that LF and HF power were measured in absolute units (ms2). Using MEG system, the units seem to �V2. Please confirm.

2. In Table 1, it is described, “Values are shown as mean ± SD”. Period is required.

3. In Line 344, it is described, “Only significant change is shown…” This may mean, “Only the results of the brain locations in which statistically significant changes were shown…”

Reviewer #3: Summary:

Thank you for the opportunity to review this interesting paper. The authors presented images of 'invisible' and visible food stimuli using a crossover design to 31 Ss and assessed LF/HF ratio and MEG activations across six frequency bands as outcome parameters. The MEG data of 14 Ss are described, with BA45 and BA47 marked as regions of interest following exploratory analyses. These differences are interpreted as evidence of " unconscious neural processes in eating behavior".

A number of concerns prevent me from recommending acceptance of the current study. Summarily, these include the authors' decision of a moving mask (unlike Takada et al. 2018, whom the authors seem to base their predictions on); lack of caloric information for food stimuli; inadequate awareness check; dropping 42% MEG data (due to participants reporting detection of at least one 'invisible' food image); no corollary measure of appetitive behavior (self-report summaries are regressed with BA45 and BA47, but this is not interpreted); and insufficient theoretical development. These points have been outlined below.

If the manuscript is revised, I recommend the authors include the 13 Ss who were dropped for potentially detecting (at least one) stimulus during invisible conditions. Awareness can be included as an ordinal factor based on the number of detections across invisible conditions. Then, if authors observe a negative association between BA activations and increasing awareness, their case for 'unconscious' processes would be strengthened.

Further points:

Introduction:

l. 58 - Discussions of how unconscious processes may influence eating behaviors, outlined in Amd and Baillet (2019) and Passarelli et al (2022), imply food-related stimuli can elicit unconscious affective and motivational responses without being coupled to propositional knowledge. This provides a potential mechanism as to how appetitive behaviors can be unconsciously moderated, which could be discussed here.

l. 70 – How was the "threshold of awareness" assessed?

l. 95 – Expand on why this onset window is notable – e.g., activations before 200-300 ms are likely to be pre-lexical, meaning they should be minimally reflective of top-down/propositional/conscious control (Amd & Baillet, 2019; Bayer et al., 2019)

Materials & Methods:

l. 153 – Both the visible and invisible conditions presented food stimuli for 33.4 ms, with the latter condition associated with a backward mask in the location of the food stimuli. Because stimulus location was held constant, how do we know response effects were not influenced by the moving mask instead of unconscious appetite-related processes? A better control would have been to increase SOAs for the visible condition or present fixation/stimulus near the bottom half of the screen followed by a mask that moves towards the bottom end of the screen. This should be described under limitations.

l. 161 – Was caloric quantity included as a covariate? Lee et al (2022) showcased high-caloric foods engage attentional processes significantly faster than low-caloric foods

l. 162 – Were the same items repeated across invisible and visible conditions? How would you account for memory effects for participants exposed to a visible->invisible sequence?

l. 167 – How was the picture "identical" to the mask? Why wasn't the actual mask used? Why were the stimulus and the (pseudo?) mask simultaneously presented (instead of a forward/backward sandwich procedure?)

l. 174 – Fixation/stimulus always appeared near the top half of the screen?

l. 279 – Who were the Ss' whose MEG data were not analyzed and why?

Results:

l. 289 – No effect of the visibility manipulation then

l. 295 – Include panel labels for visible and invisible conditions

l. 302 – Had the 13 participants who were dropped for stimulus recognition exposed to a visible>invisible sequence? A 40% attrition rate is quite high, and counts against the 'invisibility' of the food stimuli. Additionally, the awareness questionnaire was provided at the end of the task, but this may not capture conscious appraisal that was short-lived after stimulus onsets. A trial-by-trial awareness check could be more reliable and would probably increase attrition (Newell & Shanks, 2014)

l. 314 – I do not understand the point of Table 1. Are the effect sizes representative of the mean differences between participant groups whose data were analyzed vs not analyzed? I struggle to infer the relevance of (say) participants' whose data were not analyzed having higher 'emotional eating' scores relative to participants whose data were analyzed

l. 357 – Change the x-axis labels of Figure 4 to be succinct. E.g., Panel A x-label could be 'BA-45 Z-scores'. The responses for emotional eating should be standardized before regression.

Discussion:

l. 369 – you cannot make claims of 'awareness thresholds' given the limitations noted above

l. 381 – You cannot claim "could not have been" – participants may have been aware of the stimuli at the time of presentation but this had extinguished by the time of assessment.

l. 397 – why would sympathetic activity not be viewed during visible conditions then?

l. 409, 411 – Both groups showed variance in response to food stimuli above awareness thresholds. It is unclear why would there would "be a possibility" that HRV would similarly vary in the presence of food stimuli presented below awareness. One possibility is that 'partial' awareness of the food stimuli may be sufficient to generate full-scale representations (in which case observed effects would not really be reflective of 'unconscious processes' – cf. Kouider et al., 2010).

L. 439-448 – It is not clear what you're trying to say. The sentence from l. 442-448 is particularly unwieldy

l. 481 – how would indices of eating behavior influence food visibility?

l. 487 –See the discussion in Passarelli et al 2022 to extend this speculation

References:

Amd, M., & Baillet, S. (2019). Neurophysiological effects associated with subliminal conditioning of appetite motivations. Frontiers in psychology, 10, 457.

Bayer, M., Grass, A., & Schacht, A. (2019). Associated valence impacts early visual processing of letter strings: Evidence from ERPs in a cross-modal learning paradigm. Cognitive, Affective, & Behavioral Neuroscience, 19(1), 98-108.

Kouider, S., De Gardelle, V., Sackur, J., & Dupoux, E. (2010). How rich is consciousness? The partial awareness hypothesis. Trends in cognitive sciences, 14(7), 301-307.

Lee, H. H., Chien, S. E., Lin, V., & Yeh, S. L. (2022). Seeing food fast and slow: Arousing pictures and words have reverse priorities in accessing awareness. Cognition, 225, 105144.

Newell, B. R., & Shanks, D. R. (2014). Unconscious influences on decision making: A critical review. Behavioral and brain sciences, 37(1), 1-19.

Passarelli, D. A., Amd, M., de Oliveira, M. A., & de Rose, J. C. (2022). Augmenting salivation, but not evaluations, through subliminal conditioning of eating-related words. Behavioural processes, 194, 104541.

7. PLOS authors have the option to publish the peer review history of their article (what does this mean?). If published, this will include your full peer review and any attached files.

Reviewer #2: No

Reviewer #3: No

---

## [Author Response · Author response to Decision Letter 1]

28 Jun 2022

Responses to the comments

Reviewer 2

1. In Line 261, it is described that LF and HF power were measured in absolute units (ms2). Using MEG system, the units seem to �V2. Please confirm.

Response:

The unit for the R-R intervals is millisecond (ms) and thus, the unit for the power spectral density of the R-R wave intervals is ms2/Hz. Therefore, the unit for LF and HF power is ms2, as described in the manuscript (line 280, in the revised manuscript).

2. In Table 1, it is described, “Values are shown as mean ± SD”. Period is required.

Response:

Thank you for the suggestion. We revised the point suggested by the Reviewer 2 (line 340, in the revised manuscript).

3. In Line 344, it is described, “Only significant change is shown…” This may mean, “Only the results of the brain locations in which statistically significant changes were shown…”

Response:

Thank you for the suggestion. We revised the sentence to be “Only the results of the brain locations in which statistically significant changes were observed were shown” (lines 368 to 369, in the revised manuscript).

Reviewer 3

If the manuscript is revised, I recommend the authors include the 13 Ss who were dropped for potentially detecting (at least one) stimulus during invisible conditions. Awareness can be included as an ordinal factor based on the number of detections across invisible conditions. Then, if authors observe a negative association between BA activations and increasing awareness, their case for 'unconscious' processes would be strengthened.

Response:

Thank you for the suggestion. However, since our present study was not designed to determine the number of the food and/or object items each participant might have recognized in the invisible condition (i.e., They answered whether they recognized at least one image other than the mask picture after each session and therefore, the frequency they might have recognized the items was not assessed. Likewise, we could not determine whether the items they might have recognized were the food pictures or object pictures: Some participants seemed to have exclusively recognized the object pictures and some participants seemed to have exclusively recognized the food pictures, and some participants seemed to have recognized both the food and object pictures), we are not able to include the number of detections across the invisible conditions as a variable in our analysis assessing the interaction effect.

Introduction:

1) l. 58 - Discussions of how unconscious processes may influence eating behaviors, outlined in Amd and Baillet (2019) and Passarelli et al (2022), imply food-related stimuli can elicit unconscious affective and motivational responses without being coupled to propositional knowledge. This provides a potential mechanism as to how appetitive behaviors can be unconsciously moderated, which could be discussed here.

Response:

Thank you for the useful comment. We added the descriptions in the Introduction as follows: “In fact, it has been demonstrated that the greater saliva production and the increased rating of hunger were caused by coupling the food-related stimuli presented below the threshold of awareness with positively valenced terms, suggesting that the unconscious processes, which would not evoke eating-related deliberation, can effectively modulate affective and motivational responses caused by food-related stimuli (Amd et al., 2019 and Passarelli et al., 2022)” (lines 58 to 62, in the revised manuscript).

References:

1. Amd M, Baillet S. Neurophysiological Effects Associated With Subliminal Conditioning of Appetite Motivations. Frontiers in psychology. 2019;10:457. Epub 2019/03/21. doi: 10.3389/fpsyg.2019.00457. PubMed PMID: 30890986; PubMed Central PMCID: PMCPMC6411685.

2. Passarelli DA, Amd M, de Oliveira MA, de Rose JC. Augmenting salivation, but not evaluations, through subliminal conditioning of eating-related words. Behav Processes. 2022;194:104541. Epub 2021/11/24. doi: 10.1016/j.beproc.2021.104541. PubMed PMID: 34813914.

2) l. 70 – How was the "threshold of awareness" assessed?

Response:

In their study (Takada et al., 2018), it was confirmed that the participants whose data were analyzed did not recognized any images other than the mask-pictures by a questionnaire. We revised the descriptions, “Compared with a condition in which mosaic pictures created from original food stimuli were presented below the threshold of awareness, the presentation of the food stimuli below the threshold of awareness caused the activation of sympathetic nervous activity and alterations in neural activity in Brodmann’s areas (BA) 47 and 13, which are related to the alteration of sympathetic nervous activity and the level of cognitive restraint of food intake, respectively”, in the Introduction of the previous manuscript to be “Compared with a condition in which mosaic pictures created from original food stimuli were presented so as not to be recognized by the participants, the food stimuli presented so as not to be recognized by them caused the activation of sympathetic nervous activity and alterations in neural activity in Brodmann’s areas (BA) 47 and 13, which are related to the alteration of sympathetic nervous activity and the level of cognitive restraint of food intake, respectively” (lines 71 to 76, in the revised manuscript).

References:

1. Takada K, Ishii A, Matsuo T, Nakamura C, Uji M, Yoshikawa T. Neural activity induced by visual food stimuli presented out of awareness: a preliminary magnetoencephalography study. Scientific reports. 2018;8(1):3119. Epub 2018/02/17. doi: 10.1038/s41598-018-21383-0. PubMed PMID: 29449657; PubMed Central PMCID: PMC5814400.

3) l. 95 – Expand on why this onset window is notable – e.g., activations before 200-300 ms are likely to be pre-lexical, meaning they should be minimally reflective of top-down/propositional/conscious control (Amd & Baillet, 2019; Bayer et al., 2019)

Response:

Thank you for the suggestion. We added the descriptions, “In addition, it has been reported that lexical processing and thus the activation of meaning seem to start after around 200 ms of stimulus onset (Barber et al., 2007; Amd et al., 2019; Bayer et al., 2019), suggesting that the unconscious processing of the presented stimulus is performed within the time window around 0-200 ms from stimulus onset (lines 99 to 102, in the revised manuscript).

References:

1. Barber HA, Kutas M. Interplay between computational models and cognitive electrophysiology in visual word recognition. Brain Res Rev. 2007;53(1):98-123. Epub 2006/08/15. doi: 10.1016/j.brainresrev.2006.07.002. PubMed PMID: 16905196.

2. Amd M, Baillet S. Neurophysiological Effects Associated With Subliminal Conditioning of Appetite Motivations. Frontiers in psychology. 2019;10:457. Epub 2019/03/21. doi: 10.3389/fpsyg.2019.00457. PubMed PMID: 30890986; PubMed Central PMCID: PMCPMC6411685.

3. Bayer M, Grass A, Schacht A. Associated valence impacts early visual processing of letter strings: Evidence from ERPs in a cross-modal learning paradigm. Cognitive, affective & behavioral neuroscience. 2019;19(1):98-108. Epub 2018/10/21. doi: 10.3758/s13415-018-00647-2. PubMed PMID: 30341624.

Materials & Methods:

4) l. 153 – Both the visible and invisible conditions presented food stimuli for 33.4 ms, with the latter condition associated with a backward mask in the location of the food stimuli. Because stimulus location was held constant, how do we know response effects were not influenced by the moving mask instead of unconscious appetite-related processes? A better control would have been to increase SOAs for the visible condition or present fixation/stimulus near the bottom half of the screen followed by a mask that moves towards the bottom end of the screen. This should be described under limitations.

Response:

Since we thought it was suboptimal to increase the duration of the presentation of the food or object images in the visible condition in that the duration of the presentation of the food or object images in the visible condition was not the same as that in the invisible condition, which might have caused the difference in the neural responses between the conditions, we decided to use the visual presentation of which the time course of the presentation was the same as that in the invisible condition in the visible condition as described in our manuscript. However, this procedure (i.e., the procedure used in our present study) was also suboptimal in that the vertical position of the mask image in the visible condition was not the same as that in the invisible condition as the Reviewer 3 suggested; however, since we assessed the alterations in neural responses corresponding to [(food pictures in visible condition + object pictures in invisible condition) – (object pictures in visible condition + food pictures in invisible condition)] in our present study, the effect caused by the mask image was canceled out in the analysis (i.e., the contrast described above was equal to [(food pictures in visible condition – object pictures in visible condition) – (food pictures in invisible condition – object pictures in invisible condition]). We mentioned this point in the Discussion: “the vertical position of the mask image in the visible condition was not the same as that in the invisible condition because we wanted the time course of the presentation in the visible condition to be the same as that in the invisible condition; however, since we assessed the alterations in neural responses corresponding to [(food pictures in visible condition + object pictures in invisible condition) – (object pictures in visible condition + food pictures in invisible condition)] in our present study, the effect caused by the mask image was canceled out in the analysis (i.e., the contrast described above was equal to [(food pictures in visible condition – object pictures in visible condition) – (food pictures in invisible condition – object pictures in invisible condition]) and therefore, the effect caused by the difference of the position of the mask picture between the conditions seems to be minimal in our present study” (lines 530 to 540, in the revised manuscript). 

As for the suggestion made by the Reviewer 3 that “present fixation/stimulus near the bottom half of the screen followed by a mask that moves towards the bottom end of the screen” may be an alternative approach for preparing the control condition. Thank you for the suggestion.

5) l. 161 – Was caloric quantity included as a covariate? Lee et al (2022) showcased high-caloric foods engage attentional processes significantly faster than low-caloric foods

Response:

Thank you for the suggestion. The caloric quantity was not included as a covariate in our present study: For the MEG data analyses performed in our present study, it is necessary to increases the frequency of the presentation of each food item (i.e., each food item should be presented more than 50 times, considering the exclusion of the trials due to the contamination with noises) in order to assess the neural activity caused by each food item to include the caloric quantity as a covariate. We would like to assess the effect caused by the caloric quantity of each food item in the future study designed to assess the neural activity caused by each food item.

6) l. 162 – Were the same items repeated across invisible and visible conditions? How would you account for memory effects for participants exposed to a visible->invisible sequence?

Response:

The items presented in the invisible condition were the same as that presented in the visible condition; however, since the contrast corresponding to [(food pictures in visible condition – object pictures in visible condition) – (food pictures in invisible condition – object pictures in invisible condition] was analyzed in our present study as described in the manuscript (lines 530 to 540, in the revised manuscript), the memory effect such that pointed out by the Reviewer 3 were expected to be canceled out in the process of the calculation of the contrast (i.e., the possible effects caused by the visible > invisible and/or the invisible > visible sequences existed both in the food and object sessions). We added the descriptions as follows: “Likewise, although the items presented in the invisible condition were the same as that presented in the visible condition, since the contrast corresponding to [(food pictures in visible condition – object pictures in visible condition) – (food pictures in invisible condition – object pictures in invisible condition] was analyzed in our present study, the memory effect were expected to be canceled out in the process of the calculation of the contrast (i.e., the possible effects caused by the visible > invisible and/or the invisible > visible sequences existed both in the food and object sessions)” (lines 541 to 547, in the revised manuscript).

7) l. 167 – How was the picture "identical" to the mask? Why wasn't the actual mask used? Why were the stimulus and the (pseudo?) mask simultaneously presented (instead of a forward/backward sandwich procedure?)

Response:

The picture co-presented with the food and object pictures was completely the same as the mask picture (i.e., the actual mask was used). The co-presented mask picture was not an indispensable component of the visual stimuli in that the co-presented mask picture was presented both in the visible and invisible conditions and therefore, the effect caused by the co-presented picture might have been canceled out between the conditions. The reason why we added this co-presented mask picture was to attenuate the potential effect caused by the difference of the number of the recognized images between the visible and invisible conditions: On the one hand, the fixation cross, the food or object items, and the mask picture were recognized in the visible condition and the fixation and the mask picture were recognized in the invisible condition if the mask picture was not co-presented. On the other hand, the fixation cross, the food or object items, and the mask picture were recognized in the visible condition and the fixation, co-presented mask picture, and the mask picture were recognized in the invisible condition if the mask picture was co-presented. We added the sentences, “The picture co-presented with the food and object pictures was completely the same as the mask picture (i.e., the actual mask was used). The reason why we added this co-presented mask picture was to attenuate the potential effect caused by the difference of the number of the recognized images between the visible and invisible conditions: On the one hand, the fixation cross, the food or object items, and the mask picture were recognized in the visible condition and the fixation and the mask picture were recognized in the invisible condition if the mask picture was not co-presented with the food or object pictures. On the other hand, the fixation cross, the food or object items, and the mask picture were recognized in the visible condition and the fixation, co-presented mask picture, and the mask picture were recognized in the invisible condition if the mask picture was co-presented with the food or object pictures”, in the Materials and methods (lines 177 to 187, in the revised manuscript).

8) l. 174 – Fixation/stimulus always appeared near the top half of the screen?

Response:

As the Reviewer 3 suggested, the fixation and the food or object items were always presented near the bottom of the upper half of the screen. We added the sentence, “The fixation and the food or object items were always presented near the bottom of the upper half of the screen”, in the Materials and methods (lines 191 to 192, in the revised manuscript).

9) l. 279 – Who were the Ss' whose MEG data were not analyzed and why?

Response:

We added the descriptions in the Materials and methods as follows: “As described in the Results, the MEG data from 17 participants were excluded from the analysis because of the contamination of the MEG data by the magnetic noise originating from outside the shielded room, the insufficient backward masking effect, and so on” (lines 299 to 301, in the revised manuscript).

Results:

10) l. 289 – No effect of the visibility manipulation then

Response:

One of the aims of our present study was to examine whether visual food stimuli affect the central nervous system even if presented below the threshold of awareness. Although, as the Reviewer 3 suggested, the effect of visibility was not observed in our present study, we were able to show that the visual food stimuli affected the central nervous system even if presented below the threshold of awareness in our present study.

11) l. 295 – Include panel labels for visible and invisible conditions

Response:

Thank you for the suggestion. We added the labels for visible and invisible conditions in Figure 2.

12) l. 302 – Had the 13 participants who were dropped for stimulus recognition exposed to a visible>invisible sequence? A 40% attrition rate is quite high, and counts against the 'invisibility' of the food stimuli. Additionally, the awareness questionnaire was provided at the end of the task, but this may not capture conscious appraisal that was short-lived after stimulus onsets. A trial-by-trial awareness check could be more reliable and would probably increase attrition (Newell & Shanks, 2014)

Response:

Twelve participants of the 13 participants pointed out by the Reviewer 3 were dropped for stimulus recognition in the invisible condition (i.e., One participant of the 13 participants was dropped because of the failure in recognizing at least one food or object items in the visible condition): Among the 12 participants, eight participants performed a visible > invisible sequence and four participants performed an invisible > visible sequence. We added the description, “Among the 12 participants who were dropped for stimulus recognition in the invisible condition, eight participants performed a visible > invisible sequence and four participants performed a invisible > visible sequence”, in the Results (lines 326 to 329, in the revised manuscript). For the purpose of reference, the null hypothesis that there was no relationship between the sequence of the conditions (i.e., a visible > invisible or an invisible > visible) and the visibility was not rejected (χ2 test, P = 0.183).

If the participants were asked whether they recognized images other than the mask picture after each trial, the experimental time period for each participant would have been much longer than that in our present study (i.e., since the number of the trials in each session was 300 in our present study, the experimental time period for each session would be 15 min longer than our present study if participants answered whether they recognized the images other than mask picture after each trial in 3 s): Since longer experimental time period is not optimum for MEG data collection from the perspective of minimizing the body movement during the data collection and the burden of participants, we decided to ask them whether they recognized images other than the mask picture afterward. To minimize the possibility that the MEG data of the participants who in fact recognized some of the food or object items in the invisible condition were included in our analysis, we excluded the MEG data of the participants who declared that they might have seen images other than the mask picture at least one time in the invisible condition regardless of whether they were able to mention what they saw or whether the items they mentioned were actually presented in the session. We added the description on this point as follows: “In addition, since longer experimental time period is not optimum for MEG data collection from the perspective of minimizing the body movement during the data collection and the burden of participants, we decided to ask them whether they recognized images other than the mask picture after each session. To minimize the possibility that the MEG data of the participants who in fact recognized some of the food or object items in the invisible condition were included in our analysis, we excluded the MEG data of the participants who declared that they might have seen items other than the mask picture at least one time in the invisible condition regardless of whether they were able to mention what they saw or whether the items they mentioned were actually presented in the session” (lines 522 to 530, in the revised manuscript).

13) l. 314 – I do not understand the point of Table 1. Are the effect sizes representative of the mean differences between participant groups whose data were analyzed vs not analyzed? I struggle to infer the relevance of (say) participants' whose data were not analyzed having higher 'emotional eating' scores relative to participants whose data were analyzed

Response:

The effect sizes are the representative of the mean differences between participant groups whose data were analyzed vs. not analyzed. As the Reviewer 3 pointed out, although this table may suggest the possibility that individuals with high emotional eating scores have tendency of recognizing masked food pictures easier than those with low emotional eating score as described in the Discussion (lines 517 to 522, in the revised manuscript), Table 1 seems not to be directly relevant to the interpretation of our present results. If the Reviewer 3 suggest that Table 1 should be deleted from our manuscript, we would like to delete Table 1 and related discussions.

14) l. 357 – Change the x-axis labels of Figure 4 to be succinct. E.g., Panel A x-label could be 'BA-45 Z-scores'. The responses for emotional eating should be standardized before regression.

Response:

Thank you for the suggestion. We revised the x-axis labels of Figure 4 as suggested by the Reviewer 3. We used the standardized values for emotional eating and cognitive restraint in the revised Figure. We added the description, “The standardized values for the indices of emotional eating and cognitive restraint were used”, in the legend of Figure 4 (lines 390 to 391, in the revised manuscript). 

Discussion:

15) l. 369 – you cannot make claims of 'awareness thresholds' given the limitations noted above

Response:

Thank you for the suggestion. We revised the sentences to be, “In this study, participants viewed food and object pictures under visible and invisible conditions, respectively, and the neural activities over viewing the pictures were assessed by MEG”, in the Discussion (lines 393 to 395, in the revised manuscript).

16) l. 381 – You cannot claim "could not have been" – participants may have been aware of the stimuli at the time of presentation but this had extinguished by the time of assessment.

Response:

We revised the second paragraph of the Discussion as follows: “The data from individuals who answered that there was at least one picture in which they could not recognize a food item or object under the visible condition, and those from individuals who answered that there was at least one picture in which they recognized a food item or an object under the invisible condition were excluded from our analyses. Therefore, the participants whose data were analyzed reported that they have recognized food and object items under the visible condition and have not recognize food and object items under the invisible condition” (lines 404 to 410, in the revised manuscript).

17) l. 397 – why would sympathetic activity not be viewed during visible conditions then?

Response:

Thank you for the suggestion. The sentence in the previous manuscript pointed out by the Reviewer 3 was misleading. Therefore, we revised the sentence as follows: “By using the pictures of objects (i.e., not the mosaic pictures) as control stimuli in our present study, we could confirm that the increase of the sympathetic nervous activity caused by viewing the food pictures presented using the backward masking procedure reported in the previous study was not due to the difference between objects and non-objects” (lines 419 to 423, in the revised manuscript). The condition in which the food and mosaic pictures were presented without the backward masking procedure was not employed in the previous study. We mentioned the alteration of the sympathetic nervous activity caused by viewing the food pictures in the visible conditions observed in our present study in the Discussion (lines 440 to 448, in the revised manuscript).

18) l. 409, 411 – Both groups showed variance in response to food stimuli above awareness thresholds. It is unclear why would there would "be a possibility" that HRV would similarly vary in the presence of food stimuli presented below awareness. One possibility is that 'partial' awareness of the food stimuli may be sufficient to generate full-scale representations (in which case observed effects would not really be reflective of 'unconscious processes' – cf. Kouider et al., 2010).

Response:

Thank you for the suggestion. We added the descriptions, “There was no difference in the alteration of the sympathetic nervous activity caused by viewing the food pictures between the invisible and visible conditions (i.e., the main effect of conditions was not identified) in our present study, suggesting that the sympathetic nervous activity was increased by viewing the food pictures both in the visible and invisible conditions; however, since the participants whose data were analyzed reported that they did not recognize the food and object pictures exclusively in the invisible condition and the neural response to the visual food stimuli in the invisible condition was different from that in the visible condition as described below, it is speculated that the neural mechanisms by which the alteration of the sympathetic nervous activity was caused by viewing the food pictures in the invisible condition were not the same as those in the visible condition”, in the Discussion (lines 440 to 449, in the revised manuscript). 

The partial awareness hypothesis (Kouider et al., 2010) suggested by the Reviewer 3 predicts that one feature may reach consciousness while others do not and it is postulated that individuals have high confidence about the informational content at some restricted levels in the state of partial awareness. Although we are not able to exclude the possibility that the visual food stimuli of which individuals reported not to be recognized worked as the sufficient stimuli to induce the alteration of the sympathetic nervous activity in the invisible condition, since there was no evidence that some of the features of the food stimuli presented in the invisible condition reached consciousness of our participants whose data were analyzed, it may be that the alteration of the sympathetic nervous activity caused in the visible condition was related to the unconscious processing of the food stimuli in the visible condition rather than that the alteration of the sympathetic nervous activity caused in the invisible condition was due to the existence of partial awareness of the food stimuli in the invisible condition, if this is the case. We added the descriptions, “Although we are not able to exclude the possibility that the visual food stimuli of which individuals reported not to be recognized worked as the sufficient stimuli to induce the alteration of the sympathetic nervous activity in the invisible condition, since there was no evidence that some of the features of the food stimuli presented in the invisible condition reached consciousness of our participants whose data were analyzed, it may be that the alteration of the sympathetic nervous activity caused in the visible condition was related to the unconscious processing of the food stimuli in the visible condition rather than that the alteration of the sympathetic nervous activity caused in the invisible condition was due to the existence of partial awareness of the food stimuli in the invisible condition (Kouider et al., 2010), if this is the case”, in the Discussion (lines 449 to 458, in the revised manuscript).

Reference:

1. Kouider S, de Gardelle V, Sackur J, Dupoux E. How rich is consciousness? The partial awareness hypothesis. Trends Cogn Sci. 2010;14(7):301-7. Epub 2010/07/08. doi: 10.1016/j.tics.2010.04.006. PubMed PMID: 20605514.

19) l. 439-448 – It is not clear what you're trying to say. The sentence from l. 442-448 is particularly unwieldy

Response:

Thank you for the suggestion. We revised the descriptions pointed out by the Reviewer 3 as follows: “These findings indicate that it is essential to taking both the neural responses to visual food stimuli presented above the threshold of awareness and those below the threshold of awareness into consideration to understand the neural mechanisms related to eating behavior, although we could not determine the causal association between the existence of the interaction in the oscillatory brain activity and the specific features of eating behavior (i.e., emotional eating and cognitive restraint) in our present study

” (lines 481 to 487, in the revised manuscript).

20) l. 481 – how would indices of eating behavior influence food visibility?

Response:

Thank you for the suggestion. We revised the sentence pointed out by the Reviewer 3 as follows: “Our result (Table 1) may suggest that individuals with high emotional eating scores have tendency of recognizing masked food pictures easier than those with low emotional eating score; however, since this was not confirmed in our present study and the neural mechanisms by which individual differences in eating behavior affect the visibility of the masked food pictures are unknown, further study is needed on this point” (lines 517 to 522, in the revised manuscript).

21) l. 487 –See the discussion in Passarelli et al 2022 to extend this speculation

Response:

Thank you for the suggestion. We revised the descriptions regarding the speculation pointed out by the Reviewer 3 as follows: “Although our present study is in the proof of principle stage only to show the potential importance of unconscious neural processes in eating behavior, together with the reports that unconscious decision-making processes have effects on appetitive behavior mentioned in the Introduction (Amd et al., 2019 and Passarelli et al., 2022), our findings can be expected to motivate further studies seeking to clarify the neural mechanisms related to eating behavior from the perspectives of conscious and unconscious neural processes” (lines 551 to 556, in the revised manuscript). Please refer to the response to the Reviewer 3’s comment #1.

References:

1. Amd M, Baillet S. Neurophysiological Effects Associated With Subliminal Conditioning of Appetite Motivations. Frontiers in psychology. 2019;10:457. Epub 2019/03/21. doi: 10.3389/fpsyg.2019.00457. PubMed PMID: 30890986; PubMed Central PMCID: PMCPMC6411685.

2. Passarelli DA, Amd M, de Oliveira MA, de Rose JC. Augmenting salivation, but not evaluations, through subliminal conditioning of eating-related words. Behav Processes. 2022;194:104541. Epub 2021/11/24. doi: 10.1016/j.beproc.2021.104541. PubMed PMID: 34813914.

END

---

## [Decision Letter · Decision Letter 2]

23 Aug 2022

PONE-D-20-30494R2Association between eating behavior and the immediate neural activity caused by viewing food images presented in and out of awareness: a magnetoencephalography studyPLOS ONE

Dear Dr. Ishii,

Thank you for submitting your manuscript to PLOS ONE. After careful consideration, we feel that it has merit but does not fully meet PLOS ONE’s publication criteria as it currently stands. Therefore, we invite you to submit a revised version of the manuscript that addresses the points raised during the review process.

We look forward to receiving your revised manuscript.

Kind regards,

Zhishun Wang, Ph.D.

Academic Editor

PLOS ONE

Journal Requirements:

Reviewers' comments:

Reviewer's Responses to Questions

**Comments to the Author**

1. If the authors have adequately addressed your comments raised in a previous round of review and you feel that this manuscript is now acceptable for publication, you may indicate that here to bypass the “Comments to the Author” section, enter your conflict of interest statement in the “Confidential to Editor” section, and submit your "Accept" recommendation.

Reviewer #2: All comments have been addressed

Reviewer #3: All comments have been addressed

Reviewer #4: (No Response)

2. Is the manuscript technically sound, and do the data support the conclusions?

Reviewer #2: Yes

Reviewer #3: Partly

Reviewer #4: Partly

3. Has the statistical analysis been performed appropriately and rigorously? 

Reviewer #2: Yes

Reviewer #3: Yes

Reviewer #4: Yes

4. Have the authors made all data underlying the findings in their manuscript fully available?

Reviewer #2: Yes

Reviewer #3: No

Reviewer #4: Yes

5. Is the manuscript presented in an intelligible fashion and written in standard English?

Reviewer #2: Yes

Reviewer #3: No

Reviewer #4: Yes

6. Review Comments to the Author

Reviewer #2: This study was performed on healthy male volunteers who viewed pictures of food and non-food items presented both above and below the awareness threshold and the oscillatory brain activity affected by viewing the pictures was assessed by using MEG. The authors showed that neural activity corresponding to the interactions between sessions and conditions was observed in left Brodmann’s areas 45 and 47 in the high-gamma (60–200 Hz) frequency range. They demonstrated that conscious and unconscious neural processes are differently involved in eating behavior.

This paper is well written and the authors seems to adequately respond to the Reviewer s’ comments, I think that this paper is acceptable for the publication of PLOS ONE.

Reviewer #3: Thank you for the opportunity to review the revised manuscript. The revisions have made the arguments and data presentation clearer, for which I commend the authors. However, a number of earlier issues remain a concern, and prevent me from recommending acceptance of the current manuscript. If the manuscript is revised, I recommend the participants dropped for recognizing images be included as a control.

1. The exclusion of 12 participants remains a serious concern. I understand you cannot discriminate which items participants viewed, but you could still include the 12 as a control. Replicating (or not) your outcomes with the dropped sample will inform the reader of the manipulation check's validity. e.g. assuming you replicate the analyses shown in Fig 2 across the control, the awareness checks would be validated and discussions re absence of awareness vindicated. Alternatively, overlapping MEG activations between your active and control groups would suggest awareness checks were not capturing the desired construct adequately. Either finding would bolster the Results and Discussion. Dropping the participants from analysis prevents (dis)confirming those claims.

2. Given the limitations pointed out across lines 518-522, the effect sizes reported in Table 1 are still not readily interpretable - perhaps high emotional eating corresponds with greater image sensitivity overall, regardless of image content? The MEG outcomes of the control group would allow for parallel contrasts (e.g., (dis)similar activation between Invisible/Active and Control participants) and better inform the relevance of the effects.

3. Writing need to be concise. Many statements meander e.g., across lines 50-54:

"While education on diet and/or behavioral treatments for obesity are important approaches to control body weight and prevent the health problems caused by obesity [10-12],several studies have reported that regardless of these approaches, diet adherence remains low [13-15], and nearly half of patients undergoing behavioral treatment for obesity return to their original weight within 5 years of the end of the treatment [16]. "

This could be condensed to:

"Educational and/or behavioral interventions for addressing obesity are partially successful, with nearly half of treated patients returning to pre-treatment weights within 5 years of intervention completion [13-16]."

Similarly, the following passage is difficult to follow:

"On the one hand, the fixation cross, the food or object items, and the mask picture were recognized in the visible condition and the fixation and the mask picture were recognized in the invisible condition if the mask picture was not co-presented with the food or object pictures. On the other hand, the fixation cross, the food or object items, and the mask picture were recognized in the visible condition and the fixation, co-presented mask picture, and the mask picture were recognized in the invisible condition if the mask picture was co-presented with the food or object pictures."

Do you mean to say something like:

"Fixation and mask stimuli appeared at supraliminal (>34 ms) visual thresholds across visible and invisible conditions. Food/object targets appeared at supraliminal thresholds during visible conditions, and at subliminal (<34 ms) thresholds during invisible conditions"?

Additional comments:

- Head movement is a concern during MEG, and I agree that 300 trials are too much for trial-by-trial verbal reports. This could be addressed in a future work by either i) including a check at random intervals (although this will leave possibility of detection during unassessed trials...) or ii) incorporating a forced-choice awareness check after some/all trials, which would not require head movement.

Reviewer #4: 1. 170 - Food and object pictures were same in visible and invisible conditions. Even though the mean interval between two visits was approximately 1 week, participants might memorize the pictures after the visible condition visit. Since 12 participants recognized the food object pictures under the invisible condition, why not use different food and object pictures for visible and invisible conditions?

2. 329 - Was there any motion effects in the MEG data collection?

3. 353 - In figure 3 (B) and (C), was there any difference between the BA 45 and 47 in high-gamma oscillatory brain activity?

7. PLOS authors have the option to publish the peer review history of their article (what does this mean?). If published, this will include your full peer review and any attached files.

Reviewer #2: No

Reviewer #3: No

Reviewer #4: No

---

## [Author Response · Author response to Decision Letter 2]

2 Sep 2022

Responses to the comments

Reviewer 3

Thank you for the opportunity to review the revised manuscript. The revisions have made the arguments and data presentation clearer, for which I commend the authors. However, a number of earlier issues remain a concern, and prevent me from recommending acceptance of the current manuscript. If the manuscript is revised, I recommend the participants dropped for recognizing images be included as a control.

Response:

Thank you for reviewing our manuscript. As we have mentioned in the previous response to the Reviewer 3’s comment #12, we intended to minimize the possibility that the MEG data of the participants who in fact recognized some of the food or object items in the invisible condition were included in our analysis, we excluded the MEG data of the participants who declared that they might have seen images other than the mask picture at least one time in the invisible condition regardless of whether they were able to mention what they saw or whether the items they mentioned were actually presented in the session. In other words, to exclude for certain the MEG data of the participants who recognized at least one food or object item in the invisible condition, we accepted the possibility that the MEG data of the participants who did actually recognized no items in the invisible condition were excluded (i.e., the group of the participants who were dropped for recognizing images seems to include both those who recognized some of the food and/or object items in the invisible condition and those who did not actually. It is of note that the participants who were dropped for recognizing images did not necessary recognize every item in the invisible condition, if they recognized some of the items in the invisible condition). Therefore, since the comparison between the MEG data of the participants who were included in our analyses and those of the participants who were excluded from our analyses would not properly reflect the difference of the neural activity between the conscious and unconscious recognition, it is not appropriate to compare the MEG data of the participants who were included in our analyses with those of the participants who were excluded from our analyses.

In addition, the between-individual comparison that contrasts the MEG data of the participants who were included in our analyses with those of the participants who were excluded from our analyses, suggested by the Reviewer 3, may reflect the neural activity related to the individual difference in the susceptibility to the backward masking. Of course, the neural basis of the susceptibility to the backward masking is an interesting topic: however, since the aim of our present study was not focused on the susceptibility to the backward masking, we think it is not appropriate to include the participants who were dropped for recognizing images as a control in our present study, even if the participants whose MEG data were excluded certainly recognized every item in the invisible condition.

1. The exclusion of 12 participants remains a serious concern. I understand you cannot discriminate which items participants viewed, but you could still include the 12 as a control. Replicating (or not) your outcomes with the dropped sample will inform the reader of the manipulation check's validity. e.g. assuming you replicate the analyses shown in Fig 2 across the control, the awareness checks would be validated and discussions re absence of awareness vindicated. Alternatively, overlapping MEG activations between your active and control groups would suggest awareness checks were not capturing the desired construct adequately. Either finding would bolster the Results and Discussion. Dropping the participants from analysis prevents (dis)confirming those claims.

Response:

Thank you for the suggestion. As we have described above, we think it is not appropriate to include the participants who were dropped for recognizing images in our present study as a control (i.e., Since our experiment was not designed to analyze the MEG data of the participants who were dropped for recognizing images, the data from excluded participants were not comparable to those from included participants). Instead of including the factors related to the visibility of each item into the analyses as the Reviewer 3 have suggested, we carefully designed our experiment to improve the quality and reliability of our experimental data as much as possible: We concentrated on minimizing the possibility that the MEG data of the participants who recognized any of the food or object items in the invisible condition were included in our analysis and therefore, the exclusion of 12 participants was not out of the scope of our assumption.

2. Given the limitations pointed out across lines 518-522, the effect sizes reported in Table 1 are still not readily interpretable - perhaps high emotional eating corresponds with greater image sensitivity overall, regardless of image content? The MEG outcomes of the control group would allow for parallel contrasts (e.g., (dis)similar activation between Invisible/Active and Control participants) and better inform the relevance of the effects.

Response:

Thank you for the suggestion. We think it is not appropriate to include the participants who were dropped for recognizing images in our present study as a control as we have described above. In addition, we agree with the Reviewer 3 in that Table 1 in the previous manuscript was not readily interpretable and we think that Table 1 seems not to be directly related to the interpretation of our present results, we decided to delete Table 1 and the related descriptions in the revised manuscript.

3-1. Writing need to be concise. Many statements meander e.g., across lines 50-54: 

"While education on diet and/or behavioral treatments for obesity are important approaches to control body weight and prevent the health problems caused by obesity [10-12],several studies have reported that regardless of these approaches, diet adherence remains low [13-15], and nearly half of patients undergoing behavioral treatment for obesity return to their original weight within 5 years of the end of the treatment [16]. "

This could be condensed to:

"Educational and/or behavioral interventions for addressing obesity are partially successful, with nearly half of treated patients returning to pre-treatment weights within 5 years of intervention completion [13-16]."

Response:

Thank you for the suggestion. We revised the sentence in the Introduction: “Educational and/or behavioral interventions for addressing obesity are partially successful, with nearly half of treated patients returning to pre-treatment weights within 5 years of intervention completion (Harris et al., 1996; Burke er al., 1997; Nelson et al., 2002; Butryn et al., 2011)” (lines 50 to 52, in the revised manuscript).

References:

1. Harris MI. Medical care for patients with diabetes. Epidemiologic aspects. Annals of internal medicine. 1996;124(1 Pt 2):117-22. Epub 1996/01/01. doi: 10.7326/0003-4819-124-1_part_2-199601011-00007. PubMed PMID: 8554202.

2. Burke LE, Dunbar-Jacob JM, Hill MN. Compliance with cardiovascular disease prevention strategies: a review of the research. Ann Behav Med. 1997;19(3):239-63. Epub 1997/07/01. doi: 10.1007/BF02892289. PubMed PMID: 9603699.

3. Nelson KM, Reiber G, Boyko EJ, Nhanes, III. Diet and exercise among adults with type 2 diabetes: findings from the third national health and nutrition examination survey (NHANES III). Diabetes Care. 2002;25(10):1722-8. Epub 2002/09/28. doi: 10.2337/diacare.25.10.1722. PubMed PMID: 12351468.

4. Butryn ML, Webb V, Wadden TA. Behavioral treatment of obesity. Psychiatr Clin North Am. 2011;34(4):841-59. Epub 2011/11/22. doi: 10.1016/j.psc.2011.08.006. PubMed PMID: 22098808; PubMed Central PMCID: PMCPMC3233993.

3-2. Similarly, the following passage is difficult to follow:

"On the one hand, the fixation cross, the food or object items, and the mask picture were recognized in the visible condition and the fixation and the mask picture were recognized in the invisible condition if the mask picture was not co-presented with the food or object pictures. On the other hand, the fixation cross, the food or object items, and the mask picture were recognized in the visible condition and the fixation, co-presented mask picture, and the mask picture were recognized in the invisible condition if the mask picture was co-presented with the food or object pictures."

Do you mean to say something like:

"Fixation and mask stimuli appeared at supraliminal (>34 ms) visual thresholds across visible and invisible conditions. Food/object targets appeared at supraliminal thresholds during visible conditions, and at subliminal (<34 ms) thresholds during invisible conditions"?

Response:

We revised the sentences in the Materials and Methods as follows: “The participants would recognize the fixation, the food or object items, and the mask pictures in the visible condition and they would recognize the fixation and the mask picture in the invisible condition if the mask picture was not co-presented with the food or object pictures (i.e., the number of the images recognized by the participants would be three in the visible condition and that would be two in the invisible condition)” (lines 178 to 183, in the revised manuscript).

4. - Head movement is a concern during MEG, and I agree that 300 trials are too much for trial-by-trial verbal reports. This could be addressed in a future work by either i) including a check at random intervals (although this will leave possibility of detection during unassessed trials...) or ii) incorporating a forced-choice awareness check after some/all trials, which would not require head movement.

Response:

Thank you for the useful comment. We will consider the suggested approaches in our future studies.

Reviewer 4

1. 170 - Food and object pictures were same in visible and invisible conditions. Even though the mean interval between two visits was approximately 1 week, participants might memorize the pictures after the visible condition visit. Since 12 participants recognized the food object pictures under the invisible condition, why not use different food and object pictures for visible and invisible conditions?

Response:

Thank you for the comment. While using the same pictures both in the visible and invisible conditions is beneficial for adjusting the characteristics of the visual stimuli across the conditions this may be the cause of the recognition of the food and object pictures in the invisible condition if the participants memorized the pictures after the visible condition visit, as the Reviewer 4 suggested.

Among the 12 participants who were dropped for stimulus recognition, eight participants performed a visible > invisible sequence and four participants performed an invisible > visible sequence; however, since the null hypothesis that there was no relationship between the sequence of the conditions (i.e., a visible > invisible or an invisible > visible) and the visibility was not rejected (χ2 test, P = 0.183), the memory effect seems not to have been the primary cause of the recognition of the food and/or object pictures in the invisible condition. In addition, as we have mentioned in the Discussion (lines 525 to 530, in the revised manuscript), since the contrast corresponding to [(food pictures in visible condition – object pictures in visible condition) – (food pictures in invisible condition – object pictures in invisible condition] was analyzed in our present study, the memory effect was expected to be canceled out in the process of the calculation of the contrast.

2. 329 - Was there any motion effects in the MEG data collection?

Response:

As we described in the Results, the MEG data from four participants were excluded due to the contamination with magnetic noise originating from outside the shielded room. We are not able to exclude the possibility that the noise caused by the motion effects were included in the magnetic noise; however, any noise which seemed to be related to bodily motion (and that seemed to be caused by other reasons) were not observed in the MEG data included in our analysis.

3. 353 - In figure 3 (B) and (C), was there any difference between the BA 45 and 47 in high-gamma oscillatory brain activity?

Response:

Since two peaks, which were in the BA 45 and 47, were detected in the cluster in the left frontal region shown in Figure 3A in the SPM analysis and the Z-values in these peaks were similar to each other (i.e., 4.19 and 4.18, respectively), we reported the levels of increases in high-gamma oscillatory brain activity both in the BA 45 and 47 in Figure 3B and 3C, respectively. We think there are no substantial differences in the level of alterations in high-gamma oscillatory brain activity between the BA 45 and 46 (Figure 3B and 3C, and Table 1, in the revised manuscript).

END

---

## [Decision Letter · Decision Letter 3]

27 Sep 2022

Association between eating behavior and the immediate neural activity caused by viewing food images presented in and out of awareness: a magnetoencephalography study

PONE-D-20-30494R3

Dear Dr. Ishii,

We’re pleased to inform you that your manuscript has been judged scientifically suitable for publication and will be formally accepted for publication once it meets all outstanding technical requirements.

Kind regards,

Zhishun Wang, Ph.D.

Academic Editor

PLOS ONE

Additional Editor Comments (optional):

Reviewers' comments:

Reviewer's Responses to Questions

**Comments to the Author**

1. If the authors have adequately addressed your comments raised in a previous round of review and you feel that this manuscript is now acceptable for publication, you may indicate that here to bypass the “Comments to the Author” section, enter your conflict of interest statement in the “Confidential to Editor” section, and submit your "Accept" recommendation.

Reviewer #4: All comments have been addressed

2. Is the manuscript technically sound, and do the data support the conclusions?

Reviewer #4: (No Response)

3. Has the statistical analysis been performed appropriately and rigorously? 

Reviewer #4: (No Response)

4. Have the authors made all data underlying the findings in their manuscript fully available?

Reviewer #4: (No Response)

5. Is the manuscript presented in an intelligible fashion and written in standard English?

Reviewer #4: (No Response)

6. Review Comments to the Author

Reviewer #4: (No Response)

7. PLOS authors have the option to publish the peer review history of their article (what does this mean?). If published, this will include your full peer review and any attached files.

Reviewer #4: No

---

## [Editor Report · Acceptance letter]

3 Oct 2022

PONE-D-20-30494R3 

Association between eating behavior and the immediate neural activity caused by viewing food images presented in and out of awareness: a magnetoencephalography study 

Dear Dr. Ishii:

I'm pleased to inform you that your manuscript has been deemed suitable for publication in PLOS ONE. Congratulations! Your manuscript is now with our production department. 

Kind regards, 

on behalf of

Dr. Zhishun Wang 

Academic Editor

PLOS ONE